

# Snow water equivalent in the Alps as seen by gridded datasets, CMIP5 and CORDEX climate models

Silvia Terzago[1], Jost von Hardenberg[1], Elisa Palazzi[1], and Antonello Provenzale[2]

[1]Institute of Atmospheric Sciences and Climate, National Research Council of Italy, Corso Fiume 4, Torino
[2]Institute of Geosciences and Earth Resources, National Research Council of Italy, Via Moruzzi 1, Pisa

*Correspondence to:* Silvia Terzago (s.terzago@isac.cnr.it)

**Abstract.** The estimate of the current and future conditions of snow resources in mountain areas depends on the availability of reliable, high resolution, regional observation-based gridded datasets and of climate models capable of properly representing snow processes and snow-climate interactions. Owing to the sparseness of station-based reference observations, in past decades mainly passive microwave remote sensing and reanalysis products have been used to infer information on the snow water

equivalent distribution. However, the investigation has usually been limited to flat terrains as the reliability of these products in mountain areas is poorly characterized.

This work considers the available snow water equivalent datasets from remote sensing and from reanalyses for the Greater Alpine Region (GAR), and explores their ability to provide a coherent view of the snow water equivalent distribution and climatology in this area. Further we analyze the simulations from the regional and global climate models (RCMs, GCMs)

participating in the Coordinated Regional Climate Downscaling Experiment over the European domain (EURO-CORDEX) and in the latest Coupled Model Intercomparison Project (CMIP5) respectively. We evaluate their reliability in reproducing snow water equivalent against the remote sensing and reanalysis datasets previously considered.

The results of the analysis show that the distribution of snow water equivalent and the amplitude of its annual cycle are reproduced quite differently by the different remote sensing and renalysis datasets, which in fact exhibit a large spread around

the ensemble mean. We find that GCMs at spatial resolutions finer than 1.25° longitude are in closer agreement with the ensemble mean of satellite and reanalysis products in terms of RMSE and standard deviation than lower resolution GCMs. The set of regional climate models from the EURO-CORDEX ensemble provides estimates of snow water equivalent that are locally much larger than those indicated by the gridded datasets but these differences are smoothed out when snow water equivalent is spatially averaged over the Alpine domain. ERA-Interim driven RCM simulations show a snow annual cycle

comparable in amplitude to those provided by the reference datasets while GCM-driven RCMs present a large positive bias. The snow reduction expected by mid-21[st] century in the RCP 8.5 scenario is weaker in higher-resolution RCM simulations than in GCM runs.





# 1 Introduction

The increase in surface temperatures (IPCC, 2013) has relevant consequences on high elevation regions, where snow is a dominant climatic feature (Diffenbaugh et al., 2013; Barnett et al., 2005). The shift of the $0°$C isotherm to higher elevations results in a decrease in the solid-to-total precipitation ratio in low- and mid-altitude mountain areas. In addition, higher temperatures

may result in earlier snow melt and shortening of the snow cover duration. Finally, snow cover and its local-scale variability affect climate at larger scales through the snow-albedo feedback (Scherrer et al., 2012).

Changes in mountain snowpack are expected to have implications on water availability, in particular on the timing of the seasonal runoff, likely characterized in the future by earlier spring or even winter discharge and reduced flows in summer and autumn (Beniston and Stoffel, 2014; Diffenbaugh et al., 2013; Barnett et al., 2005), and on the timing of the groundwater

recharge. Similarly, changes in the seasonality and amount of winter snow cover and spring snow melt can have significant impacts on mountain ecosystems, including high-altitude vegetation (Körner, 2003) and the population dynamics of animal species that depend on snow resources (Imperio et al., 2013).

For these reasons, reliable regional estimates of current and future expected changes in snow cover are essential to develop adaptation and management strategies. Detailed studies on the recent and projected impacts of global warming in snow-

dominated regions are necessary to inform future management of water resources (Beniston and Stoffel, 2014; Stewart, 2009; Barnett et al., 2005) and to preserve essential ecosystem services for millions of people living in downstream areas. For such applications, the uncertainties associated with the future snow projections must be carefully estimated and the reliability of the model results should be assessed.

In order to evaluate the state-of-the-art Global and Regional Climate Models (GCMs, RCMs) and their future projections,

as well as to improve the representation of snow processes in such models, reliable datasets, possibly at high spatial resolution and representing the local climate characteristics in orographically complex areas, are required. However, the density of surface stations measuring snow is currently insufficient to develop a global, reliable gridded snow water equivalent dataset based on in-situ measurements, which calls for the use of alternative sources of information on snow depth and mass, derived from remote sensing observations and reanalyses (Mudryk et al., 2015).

Satellite measurements have been shown to provide a reliable picture of the global snow cover extent at relatively high spatial resolution (Brown et al., 2010; Hall and Riggs, 2007) while the estimation of snow depth and water equivalent from satellite is more challenging (Salzmann et al., 2014, see also Sect. 2). Global reanalyses provide snow water equivalent fields at horizontal resolutions that are comparable ($\sim$30 km in the zonal direction) or coarser than satellite products. Some reanalyses, such as ERA-Interim (Dee et al., 2011) and NCEP-CFSR (Saha et al., 2010), assimilate surface snow depth measurements and

satellite snow cover extent while others, such as MERRA (Rienecker et al., 2011) and 20CR (Compo et al., 2011), are not constrained by measurements and thus rely on the capability of their land-surface model component to estimate snow fields.

To date, few studies have investigated the accuracy of satellite-based and reanalysis snow water equivalent (SNW) datasets against available observations and very little is known on their performances in mountain areas. Clifford (2010), for example, compared the long-term global snow water equivalent climatology provided by the National Snow and Ice Data Center (NSIDC,



Armstrong et al., 2005), derived from passive microwave instruments, to the ERA40 reanalysis (Uppala et al., 2005) and to the output of the global climate model HadCM3 (Gordon et al., 2000; Collins et al., 2001). The largest differences between the three datasets were found for the Himalayas and for the west coast of North America, likely owing to heterogeneity of the sub-grid topography. Globally, the GCM and the reanalysis were found to be in higher agreement with each other than with the

satellite product. The GCM and reanalysis fields displayed a similar climatological annual cycle in the northern Hemisphere, a thick snow depth over Eurasia and a thin one over Siberia, while the satellite data indicated a thin snow pack in Eurasia and a thick one in Siberia overstimating snow depth with respect to the available ground observations. Another recent study by Mudryk et al. (2015) widened the analysis of Clifford (2010) by investigating additional SNW global datasets derived from satellite measurements (GlobSnow, Takala et al., 2011), from reanalyses (ERA-Interim/Land and MERRA), and from land-

surface models driven by meteorological forcing. The spread among these products was found to be lowest and their temporal correlation highest in mid-latitude boreal regions, likely owing to the fact that snow cover is generally ubiquitous during the cold season and the atmospheric circulation (midlatitude winter cyclones) is well reproduced in the models. The largest spread was found in Arctic and alpine regions, where reanalyses are poorly constrained by surface observations and the uncertainty in the meteorological forcing is higher. Alpine regions present an additional complexity due to steep elevation gradients and

sub-grid surface heterogeneities that are difficult to represent in land surface models.

The present work is devoted to review the available snow datasets, and to quantitatively assess the uncertainties in the estimation of the snow water equivalent in a alpine environment. First, we expand the study by Mudryk et al. (2015) by including additional global SNW gridded datasets obtained from remote sensing and reanalyses, and we explore how these datasets represent the snow climatology over the Greater Alpine Region (GAR). Based on this analysis, we critically discuss

the performances of state-of-the-art SNW products in an orographically complex area and we provide an estimate of the inter-dataset spread in the Alps. These results are used as a reference for evaluating the state-of-the-art climate models participating in the two major coordinated global and regional climate modeling experiments, the $5^{th}$ Coupled Model Intercomparison Project (CMIP5, Taylor et al., 2012) and the Coordinated Regional Climate Downscaling Experiment over the European domain (EURO-CORDEX, Kotlarski et al., 2014). For each model, we assess its snow water equivalent climatology against

the satellite and reanalysis datasets, and we measure the agreement among the different ensembles. The discrepancies among reanalysis and climate model simulations are discussed in relation to possible biases in the main driver of snow processes - surface air temperature and precipitation - with respect to the observational dataset E-OBS. Finally we provide an overview on the projected changes of the Alpine snow water equivalent by mid-21$^{st}$ century under a high-range emission scenario (RCP 8.5), and we highlight the differences between the CMIP5 coarse-resolution projections and the finer-resolution ones from

CORDEX.

The paper is organized as follows: section 2 introduces the datasets used for the analysis, section 3 describes the area of study, discusses the representation of orography in the current generation regional and global climate models, and summarizes the methodology employed for the data processing; section 4 reports the results in terms of (i) snowpack distribution in remote sensing products, reanalyses and climate model simulations over the Greater Alpine Region during the last decades,

(ii) inter-dataset spread in the representation of the annual cycle of snow water equivalent, and (iii) inter-dataset spread in the



representation of the snow changes expected by mid-21$^{st}$ century in the RCP8.5 scenario. Sections 5 and 6 provide a general discussion of the results in relation to other studies and conclude the paper.

## 2 Datasets

### 2.1 Remote sensing products

Satellite sensors can provide a reliable picture of the snow cover extent while the estimation of the snow water equivalent is more challenging. Passive microwave methods are based on the difference in brightness temperatures in two microwave channels, typically corresponding to frequencies of 18 GHz and 36 GHz. These methods are unable to detect very thin snow layers (i.e. less thick than 15 mm, Hancock et al., 2013) and suffer from saturation above ∼250 mm SNW (Clifford, 2010). Snow estimates from satellite are also affected by metamorphism of snow grains and snow melt: large, plate-like crystals

increase the scattering of radiation from the surface, and a shallow but dense snowpack can be misinterpreted as a thick one. Owing to its high emissivity, liquid water, either within the snowpack or at the air-snow interface, overwhelms the scattering by the snow cover and can cause underestimation of the snow thickness. Additionally, melt-refreeze processes during the melt season can cause spurious snow peak values (Hancock et al., 2013). The horizontal resolution of satellite brightness temperature measurements makes the snow estimates extremely challenging in complex terrain owing to the heterogeneity of

snow properties at subgrid scale. An eloquent example is the European Space Agency GlobSnow product in which the alpine regions are masked out because of intrinsic poorer performances and limited possibility to validate the snow estimates with surface observations (Takala et al., 2011).

Notwithstanding these limitations, satellite products are commonly used to evaluate SWE as they offer a global view on snowpack characteristics for several decades. In the present study we consider the following satellite products available for our

study area:

- Global Monthly EASE-Grid Snow Water Equivalent Climatology (Armstrong et al., 2005) provided by the National Snow and Ice Data Center (NSIDC): This dataset includes global, monthly satellite-derived snow water equivalent data from November 1978 through May 2007 at 25 km resolution (Equal-Area Scalable Earth Grid, EASE-Grid). The snow water equivalent is derived from a Scanning Multichannel Microwave Radiometer (SMMR) and selected Special Sensor

Microwave/Imagers (SSM/I).

- AMSR-E/Aqua Monthly L3 Global Snow Water Equivalent (level-3) monthly data (Tedesco et al., 2004) from the the Advanced Microwave Scanning Radiometer - Earth Observing System (AMSR-E) instrument on the NASA Earth Observing System (EOS) Aqua satellite. This dataset contains SNW data and quality assurance flags mapped to 25 km EASE-Grids from 2002 to 2011.



## 2.2 Reanalyses

A clear advantage of reanalysis products over observation-based data is that they provide global, physically-consistent estimates of all atmospheric and land-surface fields of interest, mostly constrained by observations. The reliability of reanalyses is related to the density of the assimilated observations, thus it depends on the location, the time period and the variable considered.

Reanalysis products, for example, are known to be poorly constrained by surface measurements in mountain areas where their uncertainty is larger than in other regions. Precipitation is treated differently in different reanalyses: in some cases it is a prognostic variable, i.e. it is generated by the atmospheric general circulation model and it is not constrained by observations (i.e. MERRA reanalysis, Rienecker et al., 2011); in other cases it is a prescribed forcing derived from global precipitation datasets (as in the case of CFSR and ERA-Interim/Land reanalyses). The reanalysis products considered in the present study

are:

- Climate Forecast System Reanalysis (CFSR, Saha et al., 2010) by the National Center for Environmental Prediction (NCEP), a global, high resolution, coupled atmosphere-ocean-land surface-sea ice system reanalysis, covering the period 1979-2009 and providing, among other variables, SNW fields at horizontal resolution $0.3125°$ ($\sim 38$ km at the Equator). CFSR uses two sets of observed global precipitation analyses as precipitation forcing, namely CMAP (a 5-day mean precipitation dataset at 2.5 degree latitude-longitude grid) and CPC (daily gauge analysis at 0.5 degree lat-lon over land).

CFSR snow fields are simulated by the land surface model Noah and constrained by the CFSR snow analysis. The snow analysis is based on the SNODEP model (Kopp and Kiess, 1996), which integrates surface observations, SSM/I-based detection algorithms and the NESDIS IMS North Hemisphere snow cover, based on in-situ and satellite data (Meng et al., 2012; Saha et al., 2010).

- Modern Era-Retrospective analysis for Research and Applications (MERRA, Rienecker et al., 2011) by the National Aeronautics and Space Administration (NASA), a global atmospheric reanalysis generated through the Goddard Earth Observing System Model (GEOS-5) atmospheric general circulation model and an atmospheric data assimilation system. MERRA covers the time period from 1979 through the present and it uses a grid of $1/2°$ latitude and $2/3°$ longitude with 72 vertical levels. Its land-surface model, Catchment (Koster et al., 2000), includes an intermediate complexity snow

scheme with up to three snow layers describing snow accumulation, melting, refreezing and compaction in response to meteorological forcings (Stieglitz et al., 2001).

- ERA-Interim/Land reanalysis by the European Centre for Medium-Range Weather Forecasts (ECMWF), a global re-analysis of land-surface parameters at $\sim$80 km spatial resolution covering the period 1979-2010 (Balsamo et al., 2013). ERA-Interim/Land is the result of off-line simulations performed with the improved land-surface model HTESSEL (Bal-

samo et al., 2009) forced by the meteorological fields from ERA-Interim (Dee et al., 2011) and precipitation adjustments based on GPCP v2.1. ERA-Interim/Land re-scales ERA-Interim precipitation estimates on the Global Precipitation Climatology Project (GPCP) data to remove possible biases and add the constraint of observations on a monthly time scale (Balsamo et al., 2015). In fact, in the Alps ERA-Interim/Land has been found to reduce the dry bias present in ERA-





Interim (see Appendix A for details). At large scales, the correction on snowfall has been found to be small, owing to an overall good representation in the original ERA-Interim reanalysis (Brun et al., 2013). In ERA-Interim/Land snow density and snow depth are not constrained by data assimilation owing to limited availability of surface observations. In this way the accuracy of these variables relies purely on the capability of the HTESSEL land surface model to correctly reproduce the real fields. ERA-Interim/Land has been proven to provide good quality land snow mass analyses, owing mainly to the improvements in the single layer snow scheme, with enhanced parameterizations of snow density and revised formulations for the subgrid snow cover fraction and snow albedo (Balsamo et al., 2015; Dutra et al., 2010).

– 20th Century Reanalysis version 2 (20CRv2, Compo et al., 2011) by the NOAA Earth System Research Laboratory (ESRL) Physical Sciences Division and the University of Colorado CIRES Climate Diagnostics Center, providing a synoptic-observation-based estimate of global tropospheric variability spanning the time period from 1871 to 2008. It is derived using only surface pressure observations and prescribing monthly SST and sea-ice distributions as boundary conditions for the atmosphere (Compo et al., 2011). SNW fields are available at a spatial resolution of $\sim 1.875°$ ($\sim 200$ km in the zonal direction).

### 2.3 Global climate models

Global climate models (GCMs) are the main tools available to explore climate processes and feedbacks at global scales, and to make projections in future climate change scenarios. Owing to coarse-grid limitations, current GCMs resolve explicitly only the main snow processes while the snow physics at sub-grid scale is parameterized. In such conditions, the snow schemes used in GCMs are strongly simplified: they often treat snowpack as a single-layer over the ground surface and small-scale processes such as the refreezing of melted water within the snowpack and snow metamorphism are not properly taken into account (Steger et al., 2013).

Thanks to the availability of increasing computing resources it has been possible to run models at finer and finer spatial resolutions, thus permitting a more accurate representation of the topography in orographically complex areas (Davini et al., in review; Sabin et al., 2013). Increased spatial resolution implies a more detailed view on the atmospheric forcings relevant for the mountain snowpack dynamics, i.e. altitudinal temperature gradients, precipitation distribution and phase, downward radiation, and the important physical processes could be better represented. As an example, the variable-resolution Laboratoire Meteorologie Dynamique (LMD) global climate model has been successfully employed to test the impact of the horizontal resolution on the representation of the monsoon over the South Asia (Sabin et al., 2013). They showed that the enhanced-resolution simulation at about 35 km greatly improves the representation of circulation features, the monsoon flow and the precipitation patterns with respect to the standard resolution model.

In the present study we consider the Global Climate Models included in the CMIP5 archive (http://www.cmip-pcmdi.llnl.gov/cmip5), as available in January 2015, providing the SNW variable (Table 1) during both the historical period (1850–2005) and the projection period (2006–2100) under the Representative Concentration Pathways scenario RCP8.5 (Moss et al., 2010). We consider the ensemble member r1i1p1 for all models except for EC-Earth (Hazeleger et al., 2012) for which the SNW



data were not stored in the CMIP5 archive and for which we used the ensemble member r8i1p1 run at ISAC-CNR. The spatial resolution varies from model to model in a range from 0.75° to 3.75° longitude (∼80 to 400 km in the zonal direction, see Table 1).

## 2.4 Regional climate models

Dynamical downscaling of global climate models and reanalyses through regional models can potentially provide valuable information on mountain cryosphere. Regional climate models are currently run at horizontal resolutions ranging from 50 km up to few km, allowing for a more refined representation of mountain topography and altitudinal gradients with respect to global models. Similarly to GCMs, RCMs snow schemes are strongly simplified with respect to dedicated snowpack models (Steger et al., 2013), so their main added value is to reproduce snow processes in high elevation areas, which are simply not

represented in coarse grid GCMs.

In this work we consider all the RCMs participating in the EURO-CORDEX regional climate model experiment (Kotlarski et al., 2014) and providing the snow water equivalent variable at the finest available spatial resolution, i.e. 0.11° (Table 2). We evaluate the ERAInterim-driven runs, available for 5 models at the time we downloaded the dataset in October 2016, in order to assess the RCM bias when the RCM is driven by a realistic atmospheric forcing. Three models present non-reliable

snow accumulation trends in a limited number of pixels - possibly areas masked as glaciers - so we retained only two RCMs out of the five for further investigating the historical and the future simulations under the RCP 8.5 scenario (see Section 4.1.2 for details). Specifically one, the COSMO Climate version of Local Model (CCLM, Rockel et al., 2008) provides simulations driven by several different GCMs (namely EC-Earth, CNRM-CM5, HadGEM2-ES and MPI-ESM-LR) and thus it allows for investigating the uncertainty in the snow estimate coming from the large-scale driver. The other, REMO2009, provides

simulations driven by the MPI-ESM-LR global climate model.

## 2.5 Observational datasets of air temperature and precipitation

The ability of climate models to properly reproduce snow water equivalent depends both on the accuracy of their snow schemes and on the reliability of the atmospheric forcings. Near surface air temperature (TAS) and precipitation (PR) climatologies provided by the reanalyses and the climate models considered in this study are validated against two gridded observational datasets.

Along the line of previous studies (Kotlarski et al., 2014) we consider the daily gridded dataset EOBS (version 13, Haylock et al., 2008) at 0.25° resolution, based on the European Climate Assessment and Dataset station measurements. In addition to this established and widely used reference, a second observational dataset specifically developed for the Alpine region, HISTALP (Chimani et al., 2011), is also analyzed for comparison. HISTALP allows to explore temperature and precipitation with a finer detail with respect to EOBS owing to its higher spatial resolution (0.08°).



## 3 Domain and Methods

The study domain is the Greater Alpine Region (GAR, Auer et al., 2007), extending in the range 4–19° E, 43–49° N (Fig. 1a). The complex orography of the area and the heterogeneous pattern of steep slopes and valleys make a proper representation of the climate features challenging from both an observational and a modeling point of view. As an example, Fig. 1b points out how the topography is represented in the 1-km GLOBE digital elevation model (Hastings and Dunbar, 1999), in the CORDEX regional climate models and in the CMIP5 global climate models, in terms of median and 95[th] percentile of the distribution of elevation. The median elevation is well reproduced by all models while the lowest and highest elevations are progressively cut out as the model spatial resolution decreases. While RCMs are closer to the expected values, global climate models, also the most spatially resolved ones, do not take into proper account elevations above 1500 m a.s.l. over the GAR. This limitation has to be considered when analyzing GCM outputs over mountain areas since the world reproduced by the global models has a smooth orography and simplified physical processes.

In this paper we explore the degree of agreement (i) among the reference datasets illustrated in Sect. 2.1 and 2.2, (ii) of the CORDEX and CMIP5 models compared to the reference datasets and (iii) among the different climate model ensembles, by visually inspecting the DJFMA TAS, PR, and SNW climatologies. The model performance with respect to the reference datasets is quantified using Taylor diagrams, which provide a concise statistical summary of how well patterns match a given reference in terms of their correlation (R), root-mean-square difference (RMSE), and ratio of their variances (NSD) (Taylor, 2001).

Assessments of SNW characteristics at the scale of the mountain range are obtained by spatially averaging the snow water equivalent over all areas above 1000 m a.s.l. in the GAR. To take into account the mismatch between the model topography and the real one, we use the datasets at their native resolution and weight the values by the fraction of each grid cell at elevation above 1000 m a.s.l as provided by the 1-km GLOBE (Hastings and Dunbar, 1999) digital elevation model. This procedure allows for a fair comparison between datasets characterized by different spatial resolutions, without introducing uncertainties due to regridding (for futher details see Terzago et al., 2014).

## 4 Results

### 4.1 The spatial distribution of snow water equivalent in gridded datasets

#### 4.1.1 SNW in satellite products and reanalyses

We first illustrate the spatial distribution of snow water equivalent in the satellite products and the reanalyses, hereafter referred to as the *reference* datasets, and we evaluate the differences among the reanalyses in relation to possible biases in the meteorological forcing. Figure 2 shows the multiannual mean (1980-2005) of SNW, near surface air temperature (TAS) and precipitation (PR) averaged (or accumulated in the case of PR) over the months from December to April. In order to facilitate the comparison we present the differences with respect to a given dataset: the NSIDC Global SNW Climatology for SNW,





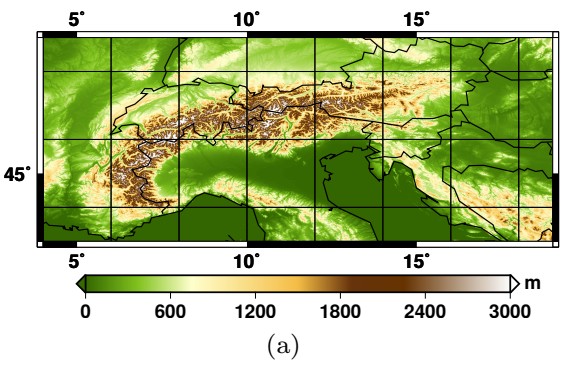

(a)

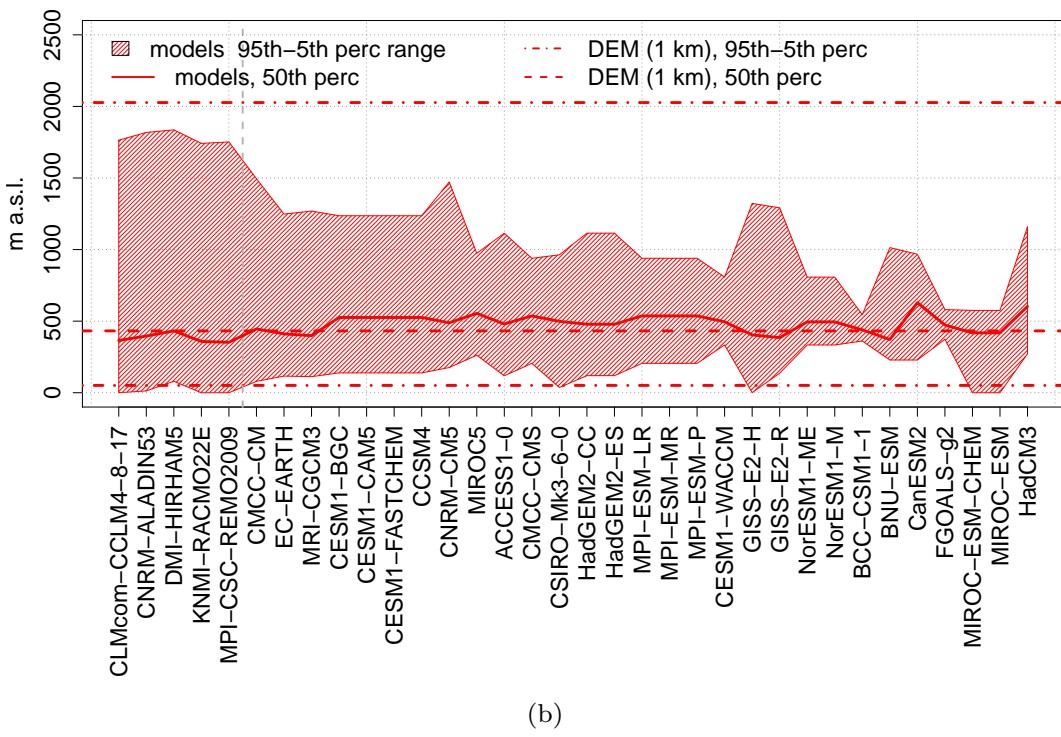

(b)

**Figure 1.** (a) Orography of the Greater Alpine Region (4-19° E; 43-49° N), as in the GLOBE 1 km digital elevation model (DEM). (b) The 95th, 50th and 5th percentiles of the elevation distribution in the DEM (dash-dot and dashed lines, respectively), compared to the corresponding values obtained from the CORDEX and CMIP5 model orographies. RCM and GCM models are ordered along the x-axis from finest to the coarsest spatial resolution. RCMs and GCMs are separated by a vertical dashed line.





since it is available for a longer period (1980-2005) than the other satellite product AMSR-E (2003-2011); EOBS observations for TAS and PR (for the latter we show percent bias).

Compared to EOBS, the alternative observational, high-resolution climatology from HISTALP (Fig. 2d-e) presents a similar temperature distribution, drier conditions at high elevations and wetter conditions at low elevations. This comparison is reported

to highlight the fact that uncertainties are greater in precipitation than temperature estimates, especially in mountain areas, and also observational datasets can exhibit biases with each other.

Focusing on snow water equivalent distribution, NSIDC Global SNW climatology (Fig. 2c) shows maximum values of about 50 $kg/m^2$ over Western Alps and 70 $kg/m^2$ over Eastern Alps. If we consider the other satellite and reanalysis products we obtain a rather heterogeneous picture. AMSR-E (Fig. 2f), which is derived as well from remote sensing observations, presents

higher values in Wester Alps and lower values in the Eastern Alps compared to the NSIDC climatology.

CFSR (Fig. 2g-i) shows TAS and PR patterns similar to EOBS over the Alpine ridge, and SNW distribution similar to NSIDC Global SNW. The similarity in SNW distribution is probably due to the fact that CFSR assimilates a snow mask derived from the same sensors as NSIDC, so these datasets are not independent.

ERA-Interim/Land (Fig. 2m-o) shows the largest SNW values, with peaks exceeding NSIDC values by more than 100

$kg/m^2$. The SNW bias is not directly explainable in terms of biases in temperature and precipitation, which indeed go towards the opposite direction (slightly warmer and drier with respect to EOBS). This result suggests that ERA-Interim/Land high SNW values are ascribable to the snow scheme in use.

The MERRA Reanalysis (Fig. 2j-l) shows thicker snowpack with respect to NSIDC Global SNW as well, especially over the Western Alps, as well as AMSR-E. The MERRA behavior can be explained by a cold bias over that area, partly compensated

by drier conditions over the Alpine peaks.

20CR (Fig. 2p-r) shows the lowest SNW values. Owing to its coarse spatial resolution, 20CR presents a warm and dry bias at high elevations and a cold and wet bias at low elevations, which result in low snow accumulation and shallow snowpack over the mountain range.

This analysis provides a quite heterogeneous picture on SNW and, despite the considerations on the biases of the drivers, it

is not possible, at the present state of knowledge, to ultimately define which product is closest to the reality over the full GAR domain. Therefore, for further analysis we use the mean of all reference datasets (Multi-Reference Mean, hereinafter MRM) calculated after interpolating all the datasets to the $0.7°$ longitude ERA-Interim/Land grid.

### 4.1.2 SNW in regional climate models

Figure 3 represents the DJFMA TAS, PR and SNW climatologies as in ERA-Interim-driven RCM simulations averaged over

the period 1990-2005. As in Fig.2 we show the biases with respect to EOBS and NSIDC SNW average fields.

All RCMs show SNW amounts several hundreds $kg/m^2$ larger than any other reference dataset (Fig. 2) at the mountain ridge and lower values at low elevations. Actually some extremely high values (shown in black) are non-reliable as they correspond to areas where snow can accumulate indefinitely, possibly areas masked as glaciers in the models. Such gridpoints present artificially high positive trends and they have to be discarded from analysis. Despite these details, RCM snow estimates are





**Figure 2.** Multiannual mean (1980-2005) of the DJFMA average (a) air temperature, (b) total precipitation and (c) snow water equivalent in the EOBS observational dataset and the NSIDC Global Monthly EASE-Grid Snow Water Equivalent Climatology respectively. Panels from (d) to (r) represent the bias of HISTALP, AMSR-E and reanalyses with respect to EOBS and NSIDC datasets, respectively.



much higher than those provided by the reference datasets, and these high values can be related to the fine representation of the orography which allows, in principle, for colder temperatures in high mountain areas, not represented in coarse-scale reanalyses, for increased solid precipitation and longer snow pack duration.

In some cases the large SNW values in RCMs can be partly explained with cold biases (RACMO22E, ALADIN53) or wet biases (HIRHAM5) with respect to the observations. In other cases (REMO2009 and CCLM4-8-17) the atmospheric forcings in correspondence of the mountain ridge are in overall agreement with observations and they do not show relevant deviations from the reference climatologies, so the differences have to be attributed to the snow scheme in use.

From the analysis of RCMs we can conclude that higher spatial resolution allows to better separate areas of snow accumulation and, consequently, to reproduce higher snow maxima in correspondence of mountain peaks.

For further investigations we will mantain only CCLM4-8-17 and REMO2009 models which present no issues in the snow accumulation trends.

### 4.1.3 SNW in global climate models

GCMs with highest spatial resolution (finer than 1.25°, Fig. 4) present considerably lower SNW amounts with respect to RCMs and comparable amounts with respect to reference datasets.

Compared to NSIDC SNW, CMCC-CM, EC-Earth and, to a smaller extent, MRI-CGCM3 and CESM1-CAM5 models, show thicker snowpack at the Northern slope of Alps and in Switzerland. A common feature of all datasets is a shallower snowpack over the Eastern Alps, at the border between Italy and Austria. This spatial pattern, characterized by an East-West gradient, with shallower snowpack in Eastern Alps and thicker snowpack in Western Alps, is resemblant to that provided by the AMSR-E satellite products rather than by NSIDC Global SNW.

BCC-CSM1-1-M, CESM1-BGC and CESM1-CAM5 show shallower snowpack than the NSIDC Global SNW, and warmer temperatures with respect to the observational datasets. In these cases the warm bias in the model can explain a less abundant snowpack.

Precipitation bias over the Alpine ridge seems comparable among the different high resolution GCMs. In fact, GCMs generally tend to a slight underestimation of winter precipitation at the ridges and to a slight overestimation at lower altitude. This uniform behavior in the precipitation pattern suggests that temperature can be the leading factor which determines biases in the estimation of the snow depth.

### 4.2 Global view on SNW products

In this section we provide a comprehensive view on all the previously considered SNW gridded datasets. The similarity of the SNW climatologies shown in Figs. 2, 3 and 4 is quantified using the metrics of Taylor diagrams (Taylor, 2001). Figure 5a compares the spatial distribution of the DJFMA snow water equivalent, averaged over the period 1980–2005, for the Multi-Reference-Mean (MRM), mean of all reference datasets to which all other datasets are compared; the Multi-Model-Mean (MMM), mean of all 35 CMIP5 models; the Multi-Model-Mean of the CMIP5 models with spatial resolution finer than 1.25° (MMM-HiRes, as in Terzago et al., 2014); the individual reference datasets; and the individual regional and global climate





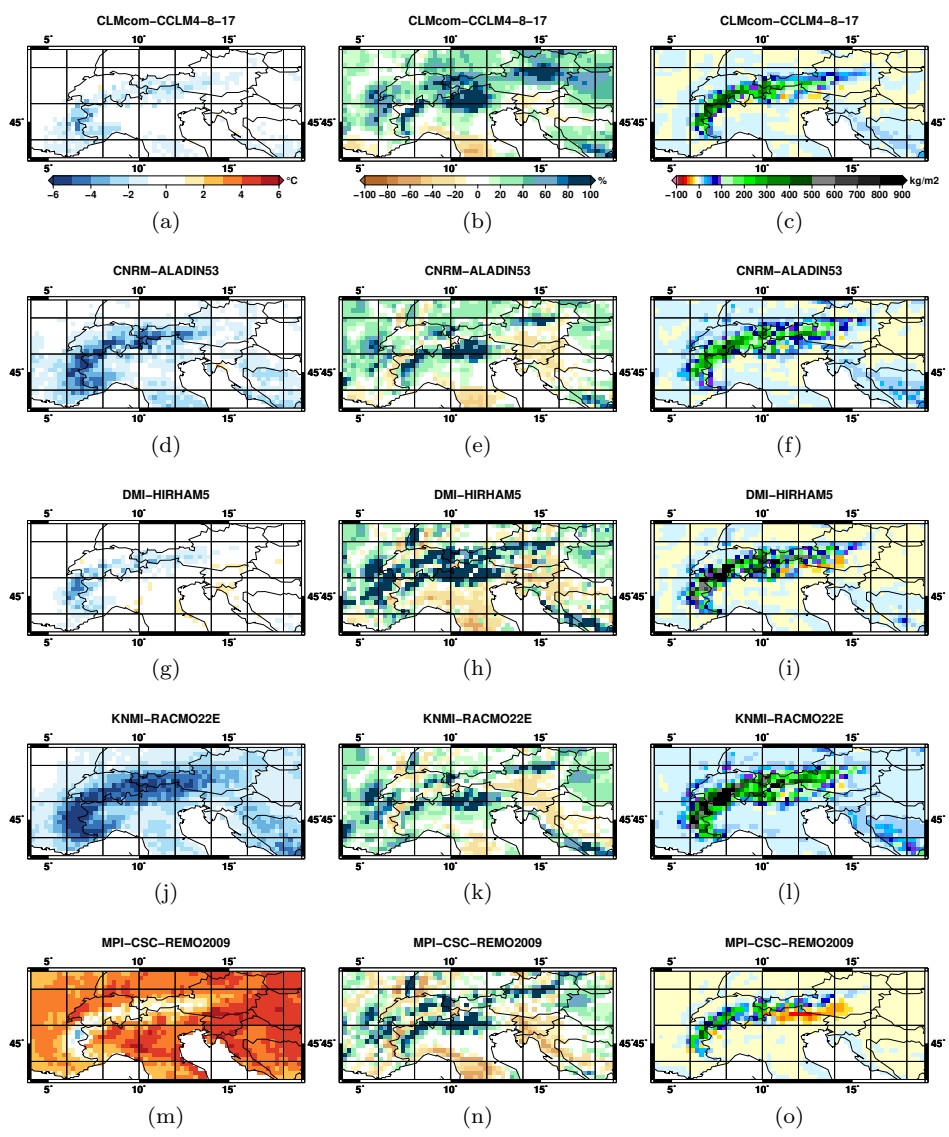

**Figure 3.** Biases in DJFMA air temperature, total precipitation and snow water equivalent in the CORDEX ERA-Interim-driven RCM simulations, with respect to EOBS and NSIDC Global Monthly EASE-Grid Snow Water Equivalent Climatologies reported in Fig. 2a,b,c.





**Figure 4.** As in Fig. 3 but for global climate models.





models. In order to compare datasets built on different coordinate reference systems and with different spatial resolutions, two different approaches have been followed. First, all remote sensing products, reanalyses and climate model outputs are reprojected onto a common grid, arbitrarily chosen as the ERA-Interim/Land 0.7° longitude grid. Alternatively, climate models are evaluated at their own resolution, comparing each model to remote sensing products and reanalyses upscaled at the climate

model grid. This second approach allows to minimize the impact of the horizontal resolution on the performances of coarse scale climate models, and in fact it is applied to GCMs only.

Figure 5a shows the results for the first approach. It provides an evaluation of the individual datasets with respect to the Multi-Reference-Mean, all resampled on the same 0.7° grid. Reference datasets are generally highly correlated with the MRM ($R > 0.85$ for all datasets except the coarsest 20CR). This feature is related to the dependence of snow water equivalent on

topography, i.e., these datasets represent larger SNW values at higher altitudes. Satellite products and the CFSR reanalysis are very close to the MRM also in terms of NSD and RMSE. The MERRA reanalysis is close to the MRM but it has larger standard deviation and a wider distribution of SNW values, compared to the MRM, satellite products and the CFSR reanalysis. The ERA-Interim/Land and 20CR reanalyses show opposite behaviors in terms of normalized standard deviation, i.e. very high and very low respectively. ERA-Interim/Land has a wider distribution of SNW values and higher SNW peaks, clearly reflected

in Fig. 2e, while 20CR has a narrow range of SNW values and a smooth SNW pattern (Fig. 2f).

Of the two RCMs considered, REMO2009 is in better agreement with the MRM in terms of RMSE and NSD. CCLM4-8-17 has large normalized standard deviation, comparable to that found for ERA-Interim/Land.

For GCMs, an important feature emerging from this analysis is that, on average, the ensemble mean of the high resolution models performs better in terms of standard deviation, root-mean-square difference and pattern correlation, with respect to the

ensemble mean of all CMIP5 GCMs. This result highlights the importance of the horizontal resolution in simulating snowpack spatial patterns (Terzago et al., 2014).

To provide a fair comparison of the GCMs and reduce the impact of the horizontal resolution on their performances, in particular on their spatial variance, each GCM is then compared to the MRM after having remapped each individual reference dataset onto the individual GCM grid, so that the reference is reshaped each time according to the model resolution. This

approach allows for a fair comparison also for low resolution models. For the sake of clarity, we present the results relative to this approach plotting separately the models with resolution *equal or finer* and *coarser* than 1.25° (Fig. 3b). The clustering based on spatial resolution reveals that coarse resolution models generally have very high or very low standard deviation (please note that the CNRM-CM5 model lays outside the range of the plot). In such circumstances the ensemble mean of the models is the result of compensating extreme behaviors. On the contrary, individual high resolution GCMs are generally closer to the

MRM and do not present extreme features: they constitute a more homogeneous ensemble that we consider for the subsequent analyses discussed below.





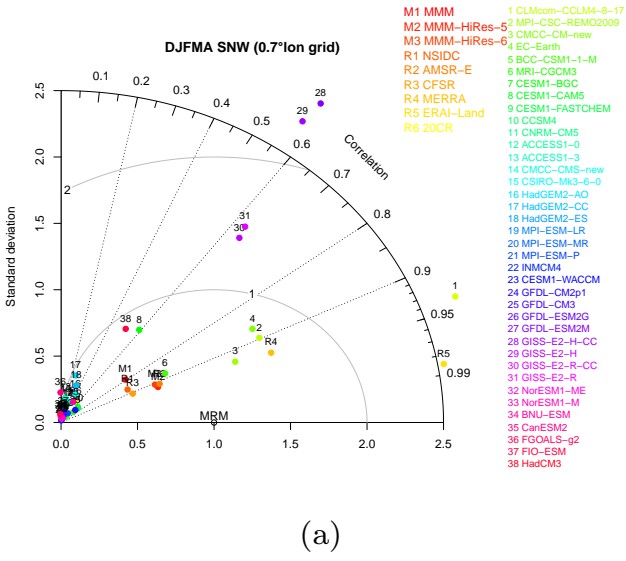

(a)

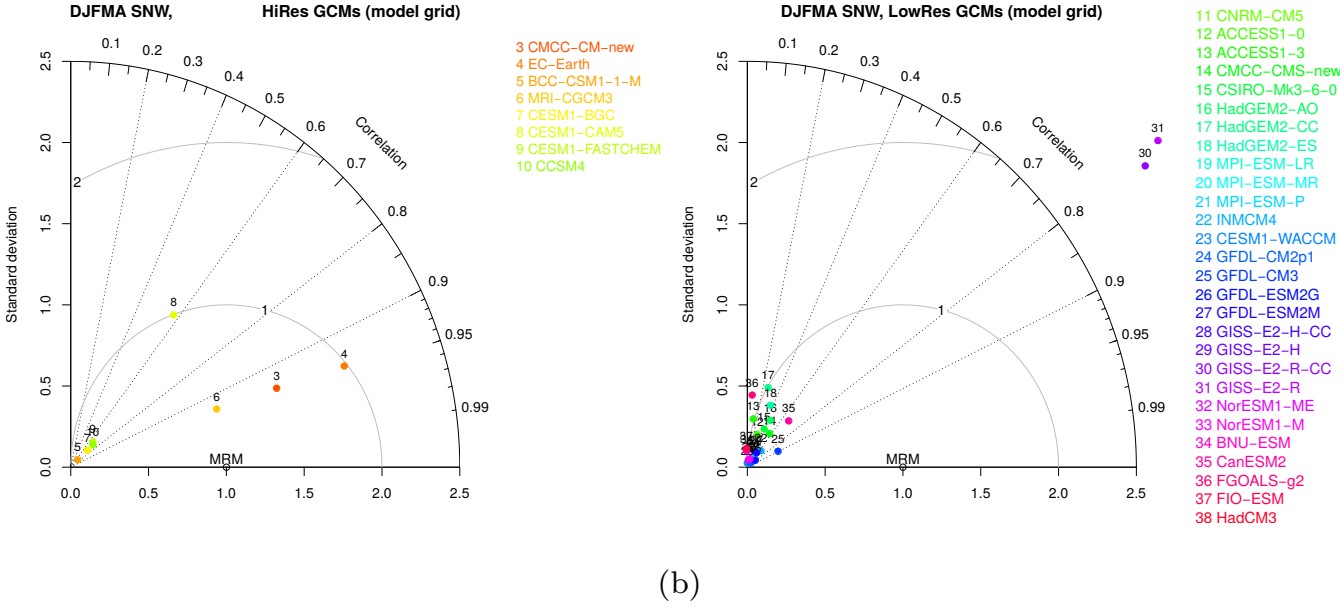

(b)

**Figure 5.** Taylor diagrams of the multiannual mean (1980-2005) of the DJFMA average snow water equivalent as described by climate models against the Multi-Reference-Mean (MRM): (a) all datasets are projected onto the same reference grid at 0.7°lon; (b) the climate models are kept at their original resolution and the reference datasets are remapped onto the grid of each model.





**Figure 6.** (a) Annual cycle of snow water equivalent in the reference datasets and (b) in CMIP5 high-resolution GCMs (spatial averages over areas above 1000 m a.s.l., temporal averages over the baseline period 1980-2005). (c) Annual cycle in ERA-Interim-driven and GCM-driven regional climate model simulations, calculated over the period 1990-2005, in comparison to reference datasets and GCM simulations.

## 4.3 Annual cycle of snow water equivalent

We show in Fig. 6a-b the annual cycle of snow water equivalent as represented by the reference datasets and by the HiRes GCMs. The monthly SNW at elevation higher than 1000 m a.s.l is spatially averaged over the Greater Alpine Region and temporally averaged over the common period 1980-2005.

The annual cycle in the reference datasets displays a unimodal distribution, with the maximum occurring in different months from Jaunary to March for different datasets. The spread in the reference datasets is quite large, ranging from 13 $kg/m^2$





SNW peak in January in the 20CR reanalysis to 150 $kg/m^2$ SNW peak in March in ERA-Interim/Land. These two products have the most extreme behavior. NSIDC and CFSR show a very similar annual cycle (and comparable spatial patterns), while MERRA presents intermediate values between these two and ERA-Interim/Land. The MRM peaks in February, at about 60 $kg/m^2$. The spread among the HiRes GCMs, although rather large, is anyway lower than that found for the reference datasets.

Snow water equivalent peaks range from 3 $kg/m^2$ according to BCC-CSM1-1-M to about 90 $kg/m^2$ according to EC-Earth. CESM1-BGC and BCC-CSM-1-1-M show very shallow SNW (few $kg/m^2$) throughout the year and a much shorter snow season, owing to a large positive bias in air temperature (Fig. 4g,m). CMCC-CM and EC-Earth present above-average values, with EC-Earth reproducing a snow cycle similar to ERA-Interim/Land, likely because they use the same land surface model, HTESSEL (Hazeleger et al., 2012). As in the case of the MRM, also the MMM-HiRes peaks in February, with comparable

but slightly lower SNW values of approximately 50 $kg/m^2$. With respect to the reference ensemble mean, the GCM ensemble mean tends to underestimate SNW throughout the snow season.

An important outcome of this analysis is that the reference datasets exhibit a large spread in the Alps, even larger than that in the high resolution GCMs. As a consequence, any assessments based on the use of individual datasets within this ensemble should be taken with extreme caution.

Figure 6c shows a synthetic view of the SNW annual cycle as in the RCMs simulations compared to reference datasets and to GCMs. ERA-Interim driven simulations provide similar results as the reference datasets. In particular the ERA-Interim-REMO2009 annual cycle is close to the ensemble mean of the reference datasets and the ERA-Interim-CCLM4-8-17 annual cycle is close to that provided by ERA-Interim/Land. Relatively larger snow water equivalent values by the CCLM4-8-17 model can be explained since this model was found to have a small cold temperature bias and a wet precipitation bias in

the DJFMA the snow season (Fig. 3a,b). The combination of colder and wetter conditions may have resulted in larger snow accumulation and duration.

GCM-driven simulations tend to overestimate the SNW annual cycle in comparison to the ERA-Interim-driven counterparts. REMO2009, when driven by MPI-ESM-LR GCM, provides SNW values close to the maximum values found in reference datasets, and CCLM-4-8-17, irrespective of the driver GCM, shows notably thicker snow pack than any reference datasets

and/or GCM. The snow peak is about 5 times higher than the reference ensemble mean, up to almost twice the ERAInterim-driven value, and it is shifted later in the snow season. Such a result reflects the biases inherent in the driving GCMs, that result in large errors in SNW estimates.

An important hint of this analysis is that despite the large differences in horizontal resolutions, GCMs and ERA-Interim-driven RCM provide comparable results in terms of SNW when the quantities are spatially averaged over the Alpine domain.

Unfortunately, conclusive statements on the accuracy of these SNW estimates, from both RCMs and GCMs, require a reliable ground truth to validate the model results.

### 4.4   Future changes in the annual cycle of SNW

Figure 7a shows the projected annual cycle of snow water equivalent by mid $21^{st}$ century (2040-2065) in the RCP8.5 scenario compared to the historical annual cycle, according to the high resolution CMIP5 models. Both the ensemble mean and the




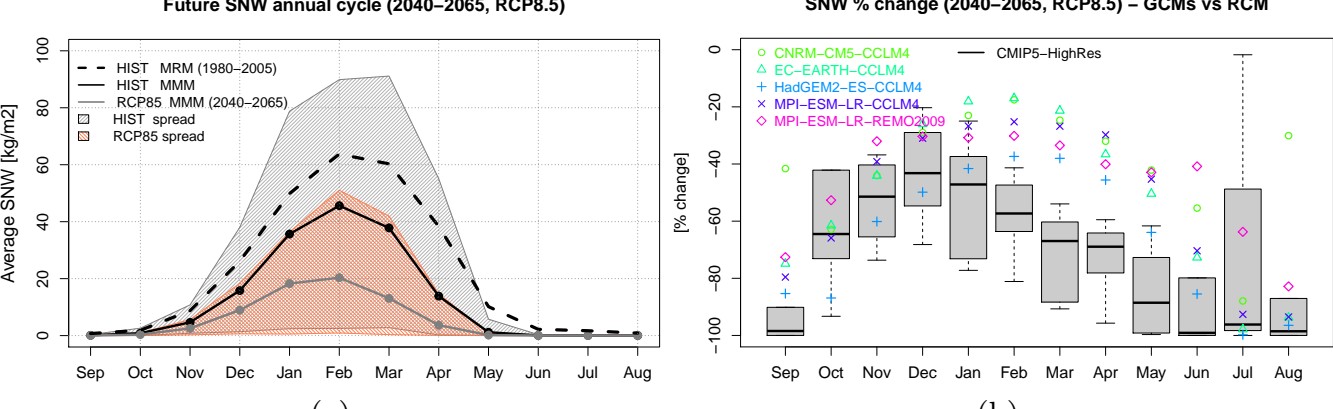

**Figure 7.** (a) Annual cycle of snow water equivalent expected by mid 21$^{st}$ century in the RCP8.5 scenario compared to the baseline 1980-2005, as provided by the Hi-Res CMIP5 models. (b) Percent change in snow water equivalent (2040-2065 average with respect to the baseline) as in the Hi-Res CMIP5 GCMs (boxplot) and RCM simulations.

spread of GCMs are shown. The SNW peak is expected to reduce by more than 50% in the future, with respect to the historical multi-model mean, reaching values of about 20 $kg/m^2$. The uncertainty on the amplitude of the snow peak is however very large and the value depends upon the selected GCM. The spread in the percent changes of SNW according to the various models (Fig. 7b) reveals the degree of inter-model consistency. The largest uncertainty is found in summer months, i.e. when

5 snow cover persists only at high altitudes and it can be very shallow. EC-Earth shows a smaller reduction while all the other models predict almost complete snow loss, on average, over the Alpine region (not shown). The lowest reduction is found in December, when the projected decrease ranges between -20% and -70% depending on the model.

For comparison we also analyze the projected changes in SNW annual cycle according to the REMO2009 model and to the CCLM4-8-17 model driven by different GCMs (Fig. 7b). Interestingly, the percent SNW reduction according to RCMs,

10 although still remarkable, is lower compared to CMIP5 GCMs. The robustness of this result should be verified by considering a larger RCM ensemble, as soon as additional RCM simulations will become available. Fig. 7b shows also the influence of the driving GCM on SNW changes. The spread among the different RCM simulations allows to evaluate the impact of the uncertainty due to the drivers of the snow changes, and its amplitude stresses the importance of performing ensemble analyses.

## 5 Discussion

15 We tested the agreement and the uncertainties of the main snow water equivalent datasets – including remote sensing products, reanalyses, global and regional climate models – in reproducing the spatial pattern and the annual cycle of snow over the Greater Alpine Region. The spatial and temporal distribution of SNW is the result of the complex interactions of temperature, precipitation, solar radiation, wind and local geographical features. In mountain areas, in particular, meteo-climatic variables



are characterized by high spatial variability depending, among other factors, on elevation, slope, aspect, and exposure to winds. The grid resolution of the remote sensing, reanalysis and climate model products is clearly insufficient to properly represent the spatial variability of snow water equivalent at small scales and at specific locations. For this reason, this study is aimed at analyzing this ensemble of largely used datasets for regional assessment, and quantifying their consistency and degree of

agreement in reproducing the average snow conditions at their own resolution.

The reference datasets provide very different pictures of the multiannual mean DJFMA snow water equivalent in the Greater Alpine Region. The satellite-derived datasets and CFSR compare better with each other than with the other products. The two satellite products are based on similar algorithms but rely on different radiometer observations, and AMSR-E doubles the spatial resolution of SMMR and SSM/I. NSIDC and CFSR are likely more similar to each other because CFSR integrates snow

analyses based on the same SSM/I observations used by snow algorithm employed in NSIDC (Meng et al., 2012). It is worth stressing that CFSR is, among the reanalyses considered in this study, the only coupled atmospheric-ocean-sea ice reanalysis; it has the highest horizontal resolution and, as ERA-Interim/Land, it is driven by observed rather than forecast precipitation fields. Interestingly, the analysis system used in CFSR for the atmosphere is similar to the one used in MERRA and despite they use almost the same input data (Saha et al., 2010) they have rather dissimilar snow water equivalent climatologies. MERRA shows

a snow distribution comparable to ERA-Interim instead, likely because they assimilate observations from the same sources and they are run at similar horizontal resolutions. MERRA compares better to the MRM in terms of normalized standard deviation and RMSE, while ERA-Interim presents higher snow values in agreement with the results obtained at Northern Hemispheric scale (Mudryk et al., 2015) and over the HKK region (Terzago et al., 2014). The ERA-Interim/Land and 20CR reanalyses show opposite behavior, i.e. very high and very low spatial variability, respectively. In particular 20CR snow water equivalent fields

are extremely smooth with respect to all other datasets. This behavior has been related to a strong warm bias in air temperature in correspondence to the alpine ridge.

The documented wide range of uncertainty has to be taken into account when using these snow datasets. Some discrepancies can be explained by possible biases in the the drivers of snow processes, the use of different land surface models, different snow schemes and different data assimilation methods, as discussed above. Additional weak points of these products are (i) their low

spatial resolution with respect to that which would be required to represent snowpack processes in mountain environments and (ii) the limited or null constraint by surface snow depth or snow water equivalent observations at high elevations (i.e., no snow assimilation). At global scale, the spread over mountain regions has been estimated to be several times larger than over midlatitude regions (Mudryk et al., 2015). Reducing this gap through improvements in the horizontal resolution and enhanced assimilation of surface data will open new perspectives for a more reliable representation of snow resources in mountain regions

at regional to global scale. Efforts have already been spent to provide a reliable atmospheric forcing, for example improving precipitation in CFSR and ERA-Interim/Land. Further inclusion of a better resolved topography allows for a more realistic representation of snow processes and could mitigate the issue of upscaling surface measurements at the model grid in the assimilation process.

GCMs have evident limitations in representing the distribution of altitudes in the Greater Alpine region, with the most

resolved models underestimating the 95[th] percentile of the distribution by 500-800 meters. GCMs do not take into proper





account the elevations above 1500 m a.s.l. which are simply non-represented in most models. On the other hand, the analysis of the CMIP5 GCMs reveals that models with spatial resolution finer or equal to 1.25° are in better agreement with the ensemble mean of the reference datasets than the whole GCM ensemble. Compared to low resolution models, the high resolution models form a more homogeneous cluster with no extreme behavior and higher score (lower RMSE and relative standard deviation

closer to one). Provided that high resolution GCMs have different characteristics and different land surface model components (Table 1), their better performance is likely due to the (relatively) finer spatial resolution. This analysis clearly indicates the added value of snow simulations at higher horizontal resolution, even for the typical resolutions of GCMs.

The influence of the single model bias with respect to the reference has been minimized by analyzing the future change in snow water equivalent with respect to the historical mean, i.e. by considering anomalies. GCM projections agree in showing a

strong reduction of snow resources by mid-21$^{st}$ century in the RCP 8.5 scenario, on average about 50% in winter and 80% in spring. The uncertainties on the amplitude of the snow water equivalent change are large, but the signal is coherent across all models.

The EURO-CORDEX regional downscaling experiment further elucidates how the horizontal resolution can affect the representation of the snow processes in mountain areas. The results from the currently available simulations at 0.11° resolution (5

ERA-Interim-driven models) show locally a much thicker average snowpack over the alpine ridge and shallower snowpack at low elevations with respect to the reference dataset. This behavior, related to the RCM finer resolution, is smothed out when snow water equivalent is spatially averaged over the Alpine domain. At regional scale, the annual cycle represented by ERA-Interim-driven RCMs results comparable to those found in the reference datasets and in GCMs. Important deviations from the reference datasets arise in GCM-driven RCM simulations, owing to the biases inherent in the GCM forcing. RCMs future

projections show weaker snow reductions with respect to the coarse scale high resolution GCMs. While few RCM models can have limited representativess of the EURO-CORDEX ensemble and a larger set of simulations has to be considered as soon as they become available, this analysis highlights the large discrepancy among the considered datasets over the historical period and calls for a reference observation-based product that could reliably represent the ground truth.

## 6   Conclusions

This study shows that the spatial and temporal distribution of snow water equivalent in the Greater Alpine Region (one of the most measured mountain regions in the world) is quite uncertain. The major available gridded snow water equivalent datasets are derived from remote sensing observations and reanalyses but they have never been properly evaluated in mountain regions owing to the limited availability of in situ snow observations. In this work, we compared such datasets to highlight the degree of agreement in the mean climatologies, to quantify their spread and assess the uncertainties associated to snow estimates. These

datasets provide very different pictures of the snow spatial distribution and seasonal cycle. Of course, mountain regions are non-optimal conditions to test these coarse-grid datasets, as surface heterogeneity at the sub-grid scale is difficult to represent for both remote sensing and reanalysis data. This argument enforces the evidence that we currently lack proper information on snowpack distribution at mountain range scale. Knowledge of the long-term variability of the snowpack at *high spatial*





*resolution* and at *mountain range* scale is limited but dramatically necessary for climate studies, for calibrating/validating models, for data assimilation in the reanalyses products and for assessing seasonal water resources. In our opinion, improving the open availability and the exchange of in-situ snow observations and developing gridded snow datasets representative of the ground truth in mountain regions is a highest priority for advancing cryospheric/hydrologic research in mountain environments.

A second action for improving snow estimates in mountain areas in both reanalyses and climate models is to pursue high resolution simulations, to allow for a better representation of the main drivers of the snow processes, i.e. temperature and precipitation patterns and their dependence on elevation. An increased horizontal resolution, and thus a more accurate representation of topography, allows for a better description of the spatial distribution and phase of precipitation and of altitudinal temperature gradients. New insights on this topic are expected by the High Resolution Model Intercomparison Project (Haarsma

et al., 2016), the CMIP6-endorsed coordinated experiment that will provide an ensemble of GCM runs at spatial resolutions significantly higher than the current generation CMIP5 models.

A further goal is the refinement of the representation of snowpack processes, that at the moment are drastically simplified, in global climate and earth system models (ESMs). This issue is being addressed by the ESM-SnowMIP initiative (van den Hurk et al., 2016, see also http://www.climate-cryosphere.org/activities/targeted/esm-snowmip) through coordinated experiments to

evaluate snow modules of large-scale climate models and quantify the required complexity to be represented in ESMs.

The present study contributes to these main challenges by providing a picture of the main available snow products and measuring the related uncertainties in the Alpine environment. The relative assessment of the capability of satellite-based products, reanalyses, RCMs and GCMs in reproducing snowpack features provides important information to both model developers and to the community of users, allowing to identify criticalities in the model components and to be aware of the strengths and limits

of the available products.

**Appendix A: ERA-Interim/Land precipitation and SNW compared to ERA-Interim**

ERA-Interim/Land and ERA-Interim snow water equivalent climatologies are derived using the ECMWF land surface model HTESSEL, being ERA-Interim/Land the result of *offline* simulation driven by meteorological forcing from the ERA-Interim atmospheric reanalysis and precipitation adjustments based on GPCP v2.1.

Percent differences of DJFMA precipitation forcing in ERA-Interim/Land (Fig. 2n) with respect to ERA-Interim in the Alpine region are reported in Fig. A1a. ERA-Interim/Land presents a larger precipitation amount over the Alpine range, partially compensating the original ERA-Interim dry bias. The additional precipitation input is reflected in a thicker snowpack, locally exceeding ERA-Interim values by more than 100 $kg/m^2$.

*Competing interests.* The authors declare that they have no conflict of interest.

*Acknowledgements.* This work has received funding from the European Union's Horizon 2020 research and innovation programme under Grant Agreements No. 641816 (CRESCENDO) and No. 641762 (ECOPOTENTIAL). This work was also supported by the Italian project of



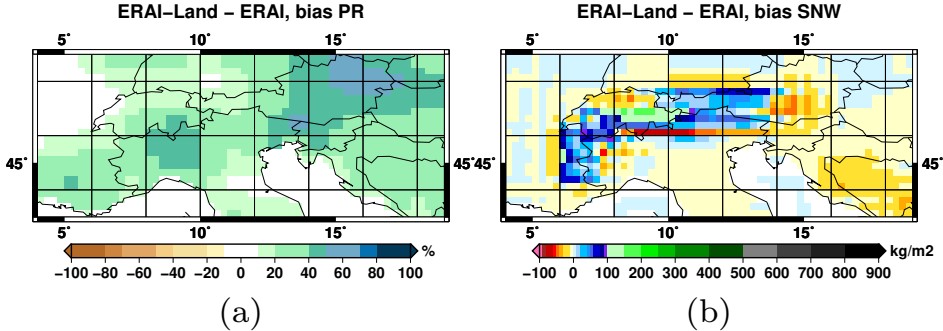

**Figure A1.** a) Percent difference in the multiannual mean (1980-2005) of the DJFMA accumulated precipitation in ERA-Interim/Land with respect to ERA-Interim; (b) Bias of ERA-Interim/Land DJFMA average snow water equivalent climatolology (1980-2005) with respect to ERA-Interim.

Interest NextData of the Italian Ministry for Education, University and Research.

We acknowledge the World Climate Research Programme's Working Group on Coupled Modelling and Working Group on Regional Climate, which are responsible for CMIP5 and CORDEX, and we thank the climate modeling groups (listed in Tables 1 and 2) for producing and making available their model output. For CMIP the U.S. Department of Energy's Program for Climate Model Diagnosis and Intercomparison

5   provides coordinating support and led development of software infrastructure in partnership with the Global Organization for Earth System Science Portals.



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




**Table 1.** Snow water equivalent datasets, including remote sensing products, reanalyses and CMIP5 Global Climate Models, used in this study. For each of these we report the land surface model (LSM, when it applies), the spatial/spectral horizontal resolution and the relevant references. CMIP5 models with horizontal resolution equal or finer than $1.25°$ lon are highlighted in bold.

| Model | Institution | LSM | Res. [°lon]/Sp.Res | Reference |
|---|---|---|---|---|
| Global SWE | National Snow and Ice Data Center | – | 25 km | Armstrong et al. (2005) |
| AMSR-E/Aqua Monthly L3 Global SWE | National Snow and Ice Data Center | – | 25 km | Tedesco et al. (2004) |
| CFSR | US National Centers for Environmental Prediction | Noah | 0.3125 | Saha et al. (2010) |
| MERRA | US National Aeronautics and Space Administration | Catchment LSM | 0.67 | Rienecker et al. (2011) |
| ERA-Interim/Land | European Centre for Medium-Range Weather Forecasts | HTESSEL | 0.7 | Balsamo et al. (2013) |
| $20^{th}$ Century Reanalysis | NOAA Earth System Research Laboratory | Noah | 1.875 | Compo et al. (2011) |
| **CMCC-CM** | Euro-Mediterranean Centre for Climate Change | ECHAM5 | **0.75** / T159 | Scoccimarro et al. (2011) |
| **EC-Earth** | EC-Earth Consortium | HTESSEL | **1.125** / T159 | Hazeleger et al. (2012) |
| **BCC-CSM1.1m** | Beijing Climate Center, China | BCC_AVIM1.0 | **1.125** / T106 | Wu et al. (2013) |
| **MRI-CGCM3** | Meteorological Research Institute, Japan | HAL | **1.125** / T159 | Yukimoto et al. (2012) |
| **CESM1-BGC** | National Center for Atmospheric Research | CLM4 | **1.25** | Hurrell et al. (2013) |
| **CESM1-CAM5** | National Center for Atmospheric Research | CLM4 | **1.25** | Hurrell et al. (2013) |
| CESM1-FASTCHEM | National Center for Atmospheric Research | CLM4 | 1.25 | Hurrell et al. (2013) |
| CCSM4 | National Center for Atmospheric Research | CLM4 | 1.25 | Gent et al. (2011) |
| CNRM-CM5 | Centre National de Recherches Météorologiques | ISBA | 1.4 / T127 | Voldoire et al. (2013) |
| ACCESS1-0 | CSIRO/BOM, Australia | MOSES2 | 1.875 / N96 | Bi et al. (2013) |
| ACCESS1-3 | CSIRO/BOM, Australia | CABLE1.0 | 1.875 / N96 | Bi et al. (2013) ?? |
| CMCC-CMS | Euro-Mediterranean Centre for Climate Change | ECHAM5 | 1.875 / T63 | Scoccimarro et al. (2011) |
| CSIRO-Mk3-6-0 | CSIRO, Australia | MOSES II | 1.875 / T63 | Collier et al. (2011) |
| HadGEM2-AO | Met Office Hadley Centre | MOSES II | 1.875 / N96 | Collins et al. (2011) |
| HadGEM2-CC | Met Office Hadley Centre | MOSES II | 1.875 / N96 | Collins et al. (2011) |
| HadGEM2-ES | Met Office Hadley Centre | MOSES II | 1.875 / N96 | Collins et al. (2011) |
| MPI-ESM-LR | Max Planck Institute for Meteorology | JSBACH | 1.875 / T63 | Giorgetta et al. (2013) |
| MPI-ESM-MR | Max Planck Institute for Meteorology | JSBACH | 1.875 / T63 | Giorgetta et al. (2013) |
| MPI-ESM-P | Max Planck Institute for Meteorology | JSBACH | 1.875 / T63 | Giorgetta et al. (2013) |
| INM-CM4 | Institute for Numerical Mathematics | INM | 2.0 | Volodin et al. (2010) |
| CESM1-WACCM | National Center for Atmospheric Research | CAM | 2.5 | Hurrell et al. (2013) |
| GFDL-CM3 | NOAA Geophysical Fluid Dynamics Laboratory | LM3 | 2.5 | Donner et al. (2011) |
| GFDL-ESM2G | NOAA Geophysical Fluid Dynamics Laboratory | LM3 | 2.5 | Dunne et al. (2012) |
| GFDL-ESM2M | NOAA Geophysical Fluid Dynamics Laboratory | LM3 | 2.5 | Dunne et al. (2012) |
| GFDL-CM2p1 | NOAA Geophysical Fluid Dynamics Laboratory | LM2 | 2.5 | Delworth et al. (2006) |
| GISS-E2-H-CC | NASA Goddard Institute for Space Studies | GISS LSM | 2.5 | Schmidt et al. (2006) |
| GISS-E2-H | NASA Goddard Institute for Space Studies | GISS LSM | 2.5 | Schmidt et al. (2006) |
| GISS-E2-R-CC | NASA Goddard Institute for Space Studies | GISS LSM | 2.5 | Schmidt et al. (2006) |
| GISS-E2-R | NASA Goddard Institute for Space Studies | GISS LSM | 2.5 | Schmidt et al. (2006) |
| NorESM1-ME | Norwegian Climate Centre | CLM4 | 2.5 | Bentsen et al. (2013) |
| NorESM1-M | Norwegian Climate Centre | CLM4 | 2.5 | Bentsen et al. (2013) |
| BNU-ESM | Beijing Normal University, China | BNU-CoLM3 | 2.8125 / T42 | [1] |
| CanESM2 | Canadian Centre for Climate Modelling and Analysis | CLASS | 2.8125 / T63 | Arora et al. (2011) |
| FGOALS-g2 | LASG/CESS, China | CLM3 | 2.8125 | Li et al. (2013) |
| FIO-ESM | The First Institute of Oceanography, China | CLM3.5 | 2.8125 / T42 | Qiao et al. (2013) |
| HadCM3 | Met Office Hadley Centre | MOSES I | 3.75 / N48 | Johns et al. (2003) |

Reference: [1]=http://esg.bnu.edu.cn/BNU_ESM_webs/htmls/index.html.



**Table 2.** EURO-CORDEX Regional Climate Models providing ERA-Interim driven runs for snow water equivalent variable at 0.11° spatial resolution considered in this study. For each of model we report also the land surface model (LSM), the number of available GCM-driven runs and the relevant references.

| Model | Institution | LSM | Ensemble members | Reference |
|---|---|---|---|---|
| CCLM4-8-17 | CLM Community | Terra-ML | 4 | Rockel et al. (2008) |
| ALADIN53 | Centre National de Recherches Météorologiques | ISBA | - | Farda et al. (2010) |
| HIRHAM5 | Danish Meteorological Institute | Hagemann (2002) | 1 | Bøssing Christensen et al. (2007) |
| RACMO22E | Royal Netherlands Meteorological Institute | HTESSEL | 2 | Van Meijgaard et al. (2012) |
| REMO2009 | Climate Service Center | Hagemann (2002) | 1 | Jacob and Podzun (1997) |