# Peer review of "Snow water equivalent in the Alps as seen by gridded datasets, CMIP5 and CORDEX climate models"

_The Cryosphere, 2016_

## Referee Comment (RC1) · Y. Cornet (Referee) · 24 Feb 2017

*Paper submitted to "The Cryosphere"*

**Snow water equivalent in the Alps as seen by gridded datasets and CMIP5 models**

Silvia Terzago[1], Jost von Hardenberg[1], Elisa Palazzi[1], and Antonello Provenzale[2]

[1]Institute of Atmospheric Sciences and Climate, National Research Council of Italy, Corso Fiume 4, Torino
[2]Institute of Geosciences and Earth Resources, National Research Council of Italy, Via Moruzzi 1, Pisa
Correspondence to: Silvia Terzago (s.terzago@isac.cnr.it)

**Reviewer: Yves Cornet, University of Liège (BE)**

**24 February 2017**

**Brief research description and my specific skills**

The authors compare the snow water equivalent content provided by 35 Global Climatic Models and 4 Regional Climatic Models to 6 reference datasets (2 products derived from microwave remote sensing) over the Great Alpine Region for historical period (1980-2005). The lack of representative field observations and the coarse resolution of those products with regard to the high heterogeneity of mountain area are declared as limitations of the analysis whose results must be interpreted with caution.

Despite this statement, however, they exploit the different models to predict the future evolution until the middle of the 21st century of the annual cycle of snow cover in this region.

In the discussion section, they mention some specificities of the models that explain the discrepancies between their results.

As I don't know anything about these models I'm not able to evaluate the truthfulness of the discussion aspect regarding the concepts and data that govern the construction of these models. But as spatial analyst I will emphasis my comments on the research paradigms that are related to these skills.

My comments, remarks and criticisms are reported below and structured in 5 sections:

- Research paradigms and hypothesis to be demonstrated,
- Methodological issues,
- Specific scientific comments,
- Comments on the document's form (text, units, figures …),
- Conclusion.

**Research paradigms and hypothesis to be demonstrated**

As recalled above, the authors are aware of the limitations of their research. The weaknesses of the product used are clearly recognized in the document. Even in this mountainous European area where a dense network of climatic stations is present in comparison to other parts of the World the number of field observations is too limited to represent fairly the spatial heterogeneity of the studied phenomenon. So the knowledge of the real wold is lacking to constrain and validate models. Field observation of snow cover can thus not be used as reference. The analysis of snow water equivalent in the Alpine region is thus a very challenging job because of spatial heterogeneity which is not taken into account in the 6 datasets used as reference (2 products derived from satellite observation and the 4 reanalysis), in the GCM and the RCM that have been compared.

Nevertheless, regarding "real world" knowledge, figure 2 shows two maps provided by the HISTALP. I tell to the authors why they didn't use this product as reference.  I also tell them why they didn't qualified HISTALP in a much more detailed way because I think it is a consistent representation of real world than the 6 ones selected as reference.

Moreover the inter-comparison of the Global and Regional climatic model and the reference datasets without knowing the "real world" evolution and its current situation is somehow disturbing for scientist. As a consequence the use of historical and predicted mean annual cycles from these models seems to me a very critical scientific paradigm which is non-pertinent.

To conclude this section, I think that the comparison between models and the so-called references is probably interesting for climatic models developers. The analysis is thus acceptable with the exception of the section dedicated to the future evolution of the snowpack. It's thus absolutely inappropriate to present it as long as the demonstration of the reliability and the realistic spatial pattern of the SNW output of the models in Mountainous regions is not made. So, the question of major interest to be answered before this predictive operation with dangerous interpretative issues is the enhanced knowledge of the snowpack from finer observations by elaborating spatially representative sampling plan of the phenomenon and developing measurement methods enabling it to be implemented.

**Methodological issues**

Weighting procedure in the computation of RMSD, normalized variance and Pearson correlation.

- This procedure is described at p. 7. You assign a weight to each grid value given by the ratio between the area above 1000 m elevation and the area of the grid cell. You should give some arguments to justify that threshold and also to convince me that it is valid whole over the GAR.
- Further in the text (p. 12 l. 3, p. 13 legend caption) you explain the procedure in another way. I think that you should correct that to remain coherent through the whole paper to avoid ambiguity.

- You should also provide a map in figure 1 () for instance with the spatial variation of this weight (area ratio).
- p. 7 l. 9-11 "This procedure allows for a fair comparison between datasets characterized by different spatial resolutions, without introducing uncertainties due to regridding". I don't totally agree with you. This procedure will enhance the importance of high area in the computation of "quality" parameters (Tailor diagrams). But high mountain zones are also very heterogeneous as low mountains zones. So the resolution difference effect will persist!!!

Interpolation. In your paper you use several words to describe the mathematical procedure used to change grid resolution (interpolation, reshaping, downscaling, remapping …). When you reduce the ground sampling distance interpolation is the right terminology.  When you degrade the resolution increasing g the ground sampling distance you perform a generalization of your geographical data and I think that this is a spatial aggregation method. You write at p. 10 : "To provide a fair comparison of the models and reduce the impact of the horizontal resolution on their performances, in particular on their spatial variance, each GCM is then compared to the MRM after having remapped each individual reference dataset onto the individual GCM grid, so that the reference is reshaped each time according to the model resolution. This approach allows for a fair comparison also for low resolution models." Despite this statement I think that you should describe in a more detailed way the methods used to "reshape" your grids giving more information about "mass conservation" condition that should be verified.

Mean (central position statistics) computation.

You compute the average of the references (satellite and Reanalysis products) .

- The number of observations (6) is very small and one of these observations is obviously an outlier (20CR). Why did you this oulier?
- This computation is performed on non- independent observation. For instance, it is quite clear for Global SWE Climatology and CFSR. So you give an exaggerated weight to those to datasets!!!

Same comments about the computation of the MMM (35 GCM or 7 HiRes GCM with ground sampling distance smaller than 1.25°).

- GCM are probably not independent (see table 1) and some are probably highly correlated and have exaggerated weight!

What is the statistical pertinence of this aggregation method to determine central position statistics? Why a mean computation to "compensating extreme behaviours" (p. 10 l. 25) ? Why don't you compute the median (for instance) in this particular case (few and not independent observations, outliers).

Absence of spatial (geographical) analysis of the differences between the various spatial grids.  To compare the grids you use 3 "quality" parameters reported in the Taylor diagrams. Even if Pearson-r is a measure of the association between variables and allows a global comparison of spatial patterns, I think that a spatial (or geographical) analysis of residuals (or the differences) is recommended to

understand the effect of spatial localization. Doing so yous should be able to improve the discussion of some climatological factors that are not integrated in the same in the same way in the models and related to air mass circulations for instance: North-South of the Alps – humid and cold air-mass flow from the North or East-West  - continentality and humid air-mass flux coming from the Adriatic towards South Eastern Alpine and Pre-alpine domain) on eventual systematic and variable biases.. This spatial analysis should thus be done for some specific and well-chosen models.

**Some specific comments**

p. 1 l. 2. I'm very surprised about your conception of high spatial resolution in this abstract and in the whole text. For remote sensors (you use satellite data) hectometric and higher ground sampling distance corresponds to low and very low spatial resolution which don't allow any description of bio-geo-physical processes on the Earth surface characterised by very high spatial frequency that are typical of mountainous area and especially the spatial variability of snow cover characteristics !

p. 1 l. 20 "The shift of the 0_C isotherm to higher elevations …"
Is it demonstrated overall on the GAR ?

p. 1 l. 22  "…decrease in the solid-to-total precipitation ratio in low- and mid-altitude mountain areas."
What do you mean by low and mid- altitude? Does that definition depends on the climatological sub-domain within the GAR ?

p.3 l. 14 What do you mean by large scale? The notion of scale in your document is somehow perturbing for cartographers and geographers that are specifically doing multiscale spatial analysis (see also p. 3 l. 14, p. 16 l. 1 for instance)! A map with a scale of 1:10000 is a large scale map that allows the representation and analysis of local physical phenomenon with small autocorrelation distance (high spatial variability). At the contrary a map with a scale of 1:1000000 is a small scale map that allows the representation and analysis of global phenomenon.

p. 6 l. 31-32 "Global climate models, also the most spatially resolved ones, do not take into proper account elevations above 1500 m a.s.l. over the GAR."
It's really a critical issue because it seems that "a very weak increasing trend towards heavier snowfalls has persisted since the 1960s" until 1999 in the Swiss Alps above the altitude of 1300 m as demonstrated by LATERNSER and SCHNEEBELI (2003, DOI: 10.1002/joc.912), for instance. But this research emphases the snow cover extent using low spatial resolution AVHRR images and you correctly state that satellite products provide a reliable picture of snow cover extent  which is not the case for snow depth or snow water equivalent (p. 2 l. 20 and 21).

p. 10 l. 2 "… arbitrarily chosen …"
This is not an acceptable. You should provide a scientific justification!

p. 10 l. 9 "… a wider distribution of SNW values, …" → "… a wider statistical dispersion of SNW values, …"
(if I understand correctly)

p.10 l. 4 "This second approach allows to minimize the impact of the horizontal resolution on the performances of GCMs."
To Be Rewritten see next comment.

p. 10 l. 18 "… reduce the impact of the horizontal resolution on their performances …"
I guess "their" refers to the models, then this sentence is not true. The impacts of the horizontal resolution on the models performance will not be reduced performing the reshaping of the reference datasets at the resolution of each GCM. To Be Rewritten.

p. 10 l. 28 "… of at least 1.25° …" → "… finer than 1.25° …"
In the document the concept of resolution is confused with that of Ground Sampling Distance!

p. 15 l. 3 "… wet precipitation bias …" Pleonasm! -→ "… overestimated precipitation …" or "… positive precipitation bias …"

p. 16 l. 23-24 "At global scale, the spread over mountain regions has been estimated to be several times larger than over midlatitude regions (Mudryk et al., 2015)."
I don't understand why you compare midlatitude regions to mountain regions. The Alps are in a midlatitude region. You should complement the qualification of midlatitude regions!

**Comments on the document's form (text, units, figures …)**

Units must be controlled:

- p. 3 l. 16 "~80° km spatial resolution" ??
- p. 8, l. 20 "$10^5$kg/m$^3$" is not consistent with the unit used to describe the SNW in the reference datasets and the GCM (figure 2 for instance → kg/m²). It seems that you did this unit conversion to compute the mean annual cycle (figure 5 p. 13)

p. 9 figure caption 2 "… with horizontal resolution higher than 1.25°." → "… with horizontal resolution finer than 1.25°"

p. 9 figure 2 "Panels (j,k) report the multiannual mean of the DJFMA accumulated snowfall derived from the HISTALP dataset. You should give a precision about the unit. I guess the unit in mm refers to the water equivalent volume per area unit! This value could be expressed using the same unit than the reference datasets and the GCM (kg/m²) assuming that 1 mm corresponds to 1 l/m² ~ 1 kg/m² !

p11. figure 3

- Labels are not readable even for points corresponding to cases with large NSD  !!
- At that scale the large amount of points near the origin must be grouped in one class with a legend identifying all the point (dates or model) in the cluster.
- Colour is not the best graphical variable to identify the signification of the point reported in the legend and it is probably not necessary. If points are grouped combine in one class and write the composition of the class. If points are dispersed then then label is sufficient!!!

**Conclusion**

To conclude my review I consider that the paper is probably interesting for model developers as long as any prediction result is performed. So I recommend the editor to publish it with several major corrections as suggested above and the removal of section 4.3 and the related comments and discussions.

---

## Referee Comment (RC2) · Anonymous Referee #2 · 5 Mar 2017

Review of "Snow water equivalent in the Alps as seen by gridded datasets, CMIP5 and CORDEX climate models" by S. Terzago, J. von Hardenberg, E. Palazzi and A. Provenzale.

Recommendation: Major revision

In this paper, the authors assess the snow water content in the Alps as represented in several atmospheric reanalyses, ERAinterim-driven (and to a lower extent CMIP5-driven) regional atmosphere models (EURO-CORDEX), and numerous CMIP5 models. I appreciate the large amount of datasets analysed in this study, however I have several concerns with the paper in its current form, and I think that a major overhaul is required before publication.

First of all, the aim of the present study is not clearly stated. Does the paper aim to

provide projections of snow water equivalent for end users (ecologists, road managers, ski resort), or does it aim to assess the models fidelity in order to point out limitations in our ability to project future snow water equivalent or for any other purpose? The aim should be better explained, and this should also be used to choose and justify which diagnostics are shown in this paper (e.g. why evaluating ERAinterin-driven RCMs in section 4.1 and 4.2 and additionally evaluating CMIP5-driven RCMs only in section 4.3?).

I also have a concern with the first diagnostics shown in this paper, i.e. the anomalies/biases represented on maps (Figs. 2-4). First, how are the datasets re-gridded prior to compute the difference? Furthermore, as the models (including reanalyses) miss the tail of the elevation distribution (as indicated in Fig.1), it is expected that they cannot account for high snowfalls observed in high-elevation areas. It seems to me that an alternative/complementary diagnostic would be to plot the snow water equivalent distribution per elevation bin. It would indicate whether the models behave well given their grid topography. I guess that the remapping used to build the Taylor diagram in Fig.5c partly addresses this, but it is not sufficient. In my opinion, this could replace sections 4.2 and 4.3 which I don't find very informative.

In addition, despite the limitations mentioned for snow water equivalent derived from passive microwave satellite observations, no attempt is made to discuss the validity of such products. The minimum would be to compare the two datasets described in section 2.1 over their common period. Of course, comparing to more datasets or to in-situ measurements would be even better. Is there any evidence that these satellite datasets are more reliable than the other datasets?

I have a possible concern with the choice of the simulations presented in the manuscript. ERAinterin-driven RCMs are more similar to AMIP models (atmospheric-only GCMs driven by observed SSTs) than to CMIP models, so in section 4.1 and 4.2, I think that comparing AMIP GCMS to ERAinterin-driven RCMs would make more sense. Then, in section 4.3 and 4.4 where the CMIP5-driven RCMs are evaluated, it

makes more sense to compare to CMIP5 models.

I have several other specific comments:

- Abstract, l.11: replace "latest" with "fifth" (in a couple of years, latest won't be clear).

- 2nd and 3rd paragraph of the Introduction: there are also concerns related to snow itself (road & airport safety, ski resorts, ...).

- Intro, l.25-26: "at relatively high spatial resolution" -> subjective, indicate a typical range.

- It would be interesting to discuss the rliability of satellite datasets in the Introduction. Note that the GlobSnow dataset is derived from satellite measurements but uses ground-based weather station data in the SWE retrieval.

- Section 2.3: Sabin et al. (2013) use LMDz as an atmosphere-only model (i.e. not coupled to an ocean), I don't know how relevant this is to the CMIP models.

- Section 2.3, last paragraph, remove "at ISAC-CNR".

- Tab. 1: there should probably be a line between satellite products and reanalyses.

- Given that LMD is mentioned, I'm surprised not to see the IPSL models in the long CMIP5 list, but anyway, there are clearly enough models in this paper.

- Sections 2.3 and 2.4: mention what kind of outputs are used (daily means or monthly means?).

- Section 2.4: what is a "non-reliable snow accumulation trends"? (and what is a reliable trend?).

- Section 2.5, about "The ability of climate models to properly reproduce snow water equivalent depends both on the accuracy of their snow schemes and on the reliability of the atmospheric forcings": it actually depends on many kinds on biases in the regional model (e.g. radiation scheme, boundary-layer scheme, etc, all being able to eventually

impact snow).

- Section 2.5: what is the interpolation method for HISTALP and EOBS?

- Section 3 could probably be merged with section 2 into a "datasets and methods" section.

- Fig.2: why showing the relative precipitation bias (in %) while the temperature and snow biases are shown as absolute errors?

- Fig.2: the caption "snow water equivalent in the EOBS observational dataset and the NSIDC Global Monthly EASE-Grid Snow Water Equivalent Climatology respectively" is misleading, it would be clearer at a first read to write that EOBS relates to (a) and (b) while NSIDC relates to (c).

- Section 4.1.1, about "In order to facilitate the comparison we present the differences with respect to a given dataset: the NSIDC Global SNW Climatology for SNW, since it is available for a longer period (1980-2005) than the other satellite product AMSR-E (2003-2011)". Ok, but it is a pity not to compare these two products aver the common period, especially given that you have claimed that "we expand the study by Mudryk et al. (2015) by including additional global SNW gridded datasets obtained from remote sensing" in the Introduction.

- Section 4.1.2: I would not say that REMO2009 is much better than the other RCMs, there is a substantial warm bias all over the domain (except maybe just over the mountain range) that could explain the relatively lower bias in SNW compared to other RCMs. Also, I would replace "CCLM4-8-17 and REMO2009 models which present no issues" with "CCLM4-8-17 and REMO2009 models which present weaker biases than other RCMs".

- Section 4.1.3: what period is used for the CMIP5 models, 1980-2005 or 1850-2005?

- Section 4.2 and its Taylor diagrams. I don't find the spatial correlation very relevant here, because it mostly relies on correlations between the topographies. Similar com-

ment for RMSE and NSD.

- Why removing the worst RCMs in section 4.2?

- I am a bit lost, why using CMIP5-driven RCMs to analyse the seasonal cycle in section 4.3 and not to evaluate the mean spatial patterns in sections 4.1 and 4.2?

- Section 4.3: I would not call 20CR a "reference dataset", it is a coarse atmospheric GCM only constrained by surface pressure and SSTs, probably more comparable to a coarse AMIP model. . .

---

## Author Comment (AC1) · 20 Apr 2017

**Reply to the comments by Yves Cornet on the manuscript "Snow water equivalent in the Alps as seen by gridded datasets, CMIP5 and CORDEX climate models"**

We thank Dr. Yves Cornet for the detailed review of the discussion paper. We have addressed all the points he raised, performing supplementary analyses that in some cases were added in the manuscript or in the new Supplementary material, hopefully improving the overall quality and clarity of the work. We noted that for few comments, the page and lines indicated by the Reviewer do not match exactly the page/lines in the manuscript published on TCD. This was not a problem as we could easily associate the comments to the correct sentences in the text.

Our point-by-point reply (black) to the suggestions and comments of the Reviewer (gray) is reported below.

**Research paradigms and hypothesis to be demonstrated**

1. "The analysis of snow water equivalent in the Alpine region is thus a very challenging job because of spatial heterogeneity which is not taken into account in the 6 datasets used as reference (2 products derived from satellite observation and the 4 reanalysis), in the GCM and the RCM that have been compared. Nevertheless, regarding "real world" knowledge, figure 2 shows two maps provided by the HISTALP. I tell to the authors why they didn't use this product as reference. I also tell them why they didn't qualified HISTALP in a much more detailed way because I think it is a consistent representation of real world than the 6 ones selected as reference."

The HISTALP gridded dataset provides a limited number of variables, including surface air temperature and total precipitation, and snowfall precipitation estimated from these. It is a good reference dataset but unfortunately it does not provide snow depth or snow water equivalent data, which are the focus of our study. This fact has been better clarified in the text, at P8 L18-20.
HISTALP temperature and precipitation climatologies have been shown in the paper and compared to the coarser resolution EOBS dataset to highlight the possible added value that high spatial resolution data can bring (the HISTALP spatial resolution is 0.083° lat/lon against 0.25° lat/lon of EOBS). Moreover, by comparing the two datasets, we highlight that uncertainties do exist *also* in *observational reference* datasets, and not only in climate models. Not unexpectedly, the uncertainty in temperature turns out to be lower than that found in precipitation. This is stated at P12 L1-4

2. "Moreover the inter-comparison of the Global and Regional climatic model and the reference datasets without knowing the "real world" evolution and its current situation is somehow disturbing for scientist. As a consequence the use of historical and predicted mean annual cycles from these models seems to me a very critical scientific paradigm which is non-pertinent. To conclude this section, I think that the comparison between models and the so-called references is probably interesting for climatic models developers. The analysis is thus acceptable with the exception of the section dedicated to the future evolution of the snowpack. It's thus absolutely inappropriate to present it as long as the demonstration of the reliability and the realistic spatial pattern of the SNW output of the models in Mountainous regions is not made. So, the question of major interest to be answered before this predictive operation with dangerous interpretative issues is the enhanced knowledge of the snowpack from finer observations by elaborating spatially representative sampling plan of the phenomenon and developing measurement methods enabling it to be implemented.

We agree that at the present state of knowledge, i.e. "without knowing the real world", projections of future snow depth are speculations. But the aim of this study is neither to state how snow water equivalent will evolve in the future nor to provide indications of the future state of snow resources. Instead we aim to show (i) how the uncertainty/spread found in the historical period project into the future, to assess the overall agreement on the relative changes with respect to each model climatology, and (ii) discuss whether the magnitude of the relative snow changes is similar in coarse and fine scale models. The spatial resolution is one order of magnitude finer in RCMs than in GCMs so that high elevation areas are resolved better in the former. We had two main questions in mind: (1) how does the resolution affect snow depth representation and its future changes in the Alpine environment? And, (2) is there any specific feature emerging in higher resolution projections, or are they indistinguishable from the lower resolution ones? This investigation is corroborated by a recent study comparing "bias corrected" and "non-bias corrected" snowfall projections of EURO-CORDEX RCM models (Frei et al. 2017). In that study bias corrected RCM snowfall was constrained to a snowfall reference dataset derived from 2 km resolution gridded temperature and precipitation data. According to that analysis, the relative change (RCP8.5 vs baseline) of the mean September-May snowfall is comparable whether applying or not the bias correction and the bias adjustment does not seem to have any significant effect on the trend.

We added in the introduction (P4 L1–6) a sentence that states in a clearer way what the main purposes of this study are.

**Methodological issues**

Weighting procedure in the computation of RMSD, normalized variance and Pearson correlation.

- - This procedure is described at p. 7. You assign a weight to each grid value given by the ratio between the area above 1000 m elevation and the area of the grid cell. You should give some arguments to justify that threshold and also to convince me that it is valid whole over the GAR.
  - Further in the text (p. 12 l. 3, p. 13 legend caption) you explain the procedure in another way. I think that you should correct that to remain coherent through the whole paper to avoid ambiguity.

The weighting procedure mentioned by the reviewer in the first item above is applied only when snow water equivalent fields are spatially averaged over the Greater Alpine Region, i.e. in the plots shown in Figures 6 and 7, and not in the Taylor diagrams. The spatial averages of Figures 6 and 7 are intended to be representative of the mountains only, so we exclude the areas below 1000 m a.s.l. (we recognize that this threshold is arbitrary but we think that it could be appropriate in the GAR for focusing on high-altitude regions only) using this weighting procedure.  The detail of the procedure is as follows: we weigh the snow water equivalent values at each grid cell by the area of the grid cell with mean elevation higher than 1000 m a.s.l. using a Digital Elevation Model at high spatial resolution (1 km), then the weighted values are spatially averaged over the domain of interest, the Greater Alpine Region. This is better explained in the manuscript at P10 L13-19.

For the Taylor diagrams, we calculated the root mean square error (RMSE), normalized standard deviation (NSD) and the correlation coefficient (R) over the full domain, without applying any weighting based on elevation.  In this case the multiannual mean snow water equivalent was simply remapped onto the target grid conserving the snow mass from the original and the target grid cells. To this end we used a standard function incorporated in the CDO (Climate Data Operators) software mostly used in the climate community to handle climate model data in netCDF format. The CDO "remapcon" function performs an area-weighted remapping where the interpolation weights are based on the fractional area overlap of the original and target grid cells, following Jones, 1999. This methodology has been explained at P10 L6-12.

Jones, P.W. 1999, *Monthly Weather Review*, **127**, 2204-2210)

- - You should also provide a map in figure 1 () for instance with the spatial variation of this weight (area ratio).

Actually the "map of weights" is resolution-dependent so we should provide a map of weights for each dataset considered in this study. These maps would show for each coarse scale grid cell "the fraction of the area of the grid cell with mean elevation higher than 1000 m", where the topography is taken from the GLOBE Digital Elevation Model at 1 km resolution. This procedure has been better explained at P10 L13-19. We report below (Fig R1) two examples of maps of weights, the former referring to the CFSR reanalysis and the latter to the EC-Earth GCM. We prefer to not provide the maps of weights for each model in the manuscript because we think that this level of detail is too high for the general purpose of the paper.

[Figure]

Fig. R1. "Map of weights" showing the fraction of each grid cell at elevation above 1000 m a.s.l. for the CFSR reanalysis and the global climate model EC-Earth. The reference topography is taken from the 1 km digital elevation model GLOBE (Hastings and Dunbar, 1999).

- - p. 7 l. 9-11 "This procedure allows for a fair comparison between datasets characterized by different spatial resolutions, without introducing uncertainties due to regridding". I don't totally agree with you. This procedure will enhance the importance of high area in the computation of "quality" parameters (Tailor diagrams). But high mountain zones are also very heterogeneous as low mountains zones. So the resolution difference effect will persist!!!

This comments reveals a misunderstanding. As explained before the weights are not used to calculate quality parameters (Taylor diagrams) for which we use the full domain. Instead, we used the weights approach to spatially average datasets characterized by very different spatial resolutions over the same domain (GAR above 1000 m a.s.l. ), without interpolating the model data. The sentence has been rephrased in the manuscript at P10 L9-11.

Interpolation. In your paper you use several words to describe the mathematical procedure used to change grid resolution (interpolation, reshaping, downscaling, remapping ...). When you reduce the ground sampling distance interpolation is the right terminology. When you degrade the resolution increasing g the ground sampling distance you perform a generalization of your geographical data and I think that this is a spatial aggregation method. You write at p. 10 : "To provide a fair comparison of the models and reduce the impact of the horizontal resolution on their performances, in particular on their spatial variance, each GCM is then compared to the MRM after having remapped each individual reference dataset onto the individual GCM grid, so that the reference is reshaped each time according to the model resolution. This approach allows for a fair comparison also for low resolution models." Despite this statement I think that you should describe in a more detailed way the methods used to "reshape" your grids giving more information about "mass conservation" condition that should be verified.

We remapped the six reference datasets onto each climate model grid using a conservative remapping, in detail the CDO remapcon function (CDO 2015). This function performs an area-weighted remapping, where

the interpolation weights are based on the fractional area overlap of the source and the destination grid cells, following Jones, 1999. Such interpolation weights applied to the source field allow to conserve the fluxes or water budgets from the source to the destination grid.
This procedure has been better explained adding some details at P10 L18-19.

CDO 2015: Climate Data Operators.  Available at: http://www.mpimet.mpg.de/cdo

Mean (central position statistics) computation.
You compute the average of the references (satellite and Reanalysis products) .

- - The number of observations (6) is very small and one of these observations is obviously an outlier (20CR). Why did you this oulier?

Thank you for this comment. We agree that after the assessment of its poor performance in representing SNW climatology,  the 20CR reanalysis should not be considered as a "reference". We repeated all the analysis excluding the 20CR from the Multi-Reference-Mean, and consequently figures 5,6 and 7 have been updated in the main text. The new procedure is explained at P12 L31-34;

- This computation is performed on non- independent observation. For instance, it is quite clear for Global SWE Climatology and CFSR. So you give an exaggerated weight to those to datasets!!!

The interdependency of the Global SWE Climatology and the CFSR snow outputs has been better clarified in the text (P6 L2-3; P12 L10-17 and following). Both products integrate, but to different extents, the Special Sensor Microwave Imager (SSM/I) data. The Global SWE Climatology is specifically derived from Special Sensor Microwave Imager (SSM/I) data. The CFSR snow output is mainly based on the Noah land surface model first guess, and a daily snow analysis based on several inputs - among others Special Sensor Microwave Imager (SSM/I) data – is used to constrain the model first guess (Meng et al., 2012). In detail, CFSR snow depth/SWE are limited in the upper and lower boundaries by the snow analysis (it cannot be larger than twice and lower than half the snow analysis) but the temporal evolution of snow depth and SWE is determined by the Noah model. In conclusion, as the similarity between the two datasets is in the similar range of variability, we decided to include both.

Same comments about the computation of the MMM (35 GCM or 7 HiRes GCM with ground sampling distance smaller than 1.25°).

- GCM are probably not independent (see table 1) and some are probably highly correlated and have exaggerated weight!

We totally agree that the climate models are not independent from one another, and several previous studies (e.g. Knutti et al., 2013; Sanderson et al., 2015) focus on this issue. For example in Figure R2 below, now included in the Supplementary Material as Fig. S02, we report the spatial distribution of the DJFMA SNW in the 8 GCMs with horizontal resolution not coarser than 1.25°, referred to as "high-resolution" HiRes GCMs in the manuscript. These high-resolution models are CMCC-CM, EC-Earth, MRI-CGCM3, BCC-CSM1-1-M and four models from the CESM-family. Out of the four CESM-family models, one, CESM1-CAM5, shows a distinct behaviour. The other three (CESM1-BGC, CESM1-FASTCHEM and CCSM4) present very similar SNW patterns (Figure R2, last row), and similar RMSE, NSD and correlations values (Figure 5b in the main text). In order to have a model ensemble including models as independent as possible, we consider in the ensemble mean (MMM-HiRes) only one out of the three (CESM1-BGC). The analyses of figures 6 and 7 are then based on 6 high-resolution models, namely CMCC-CM, EC-Earth, MRI-CGCM3, BCC-CSM1-1-M, CESM1-CAM5 and CESM1-BGC.

In Figure 5a MMM-HiRes refers to the multi-model-mean of these 6 models. This choice has been explained in text at P13 L5-8 and later on at P18 L14-22.

Concerning GCMs with spatial resolution coarser than 1.25° it is difficult to evaluate their degree of interdependence from the Taylor diagrams. However, owing to their overall poor performances in the representation of SNW, and not being the focus of the paper, the aspect of their interdependency is not investigated further (explained in the main text at P18 L20-22).

[Figure]

Fig R2: Multiannual mean (1980-2005) snow water equivalent in the GCMs with spatial resolution finer or equal to 1.25°.

What is the statistical pertinence of this aggregation method to determine central position statistics? Why a mean computation to "compensating extreme behaviours" (p. 10 l. 25) ? Why don't you compute the median (for instance) in this particular case (few and not independent observations, outliers).

We agree that it is interesting to explore the case in which the median is used as metrics, instead of the mean. In order to address this comment we explored two different approaches:

1) We considered as "reference" the median of the 5 datasets (NSIDC, CFSR, MERRA, ERAI/Land and 20CR), we calculated the median of CMIP5 models (full ensemble and HIRES ensemble) and we repeated the analysis of the Taylor diagram. The results are shown in Figure R3 . The median is shifted towards very small SNW values as NSIDC, CFSR and 20CR provide low snow. Consistently the normalized standard deviation of the MERRA reanalysis exceeds 2.5 and that of ERA-Interim lies outside the range of the plot, as well as for many climate models.
2) we consider as "reference" the average of 4 datasets (NSIDC, CFSR, MERRA, ERAI/Land), hence excluding the 20CR reanalysis from the "reference" statistics, as suggested by both reviewers. In this case we have a well-balanced ensemble, with NSIDC, CFSR showing low snow and MERRA, ERAI/Land showing large snow amounts. The results are shown in the new Figure 5a of the revised manuscript.

Considering the results of the two approaches we think that this second metric is the most appropriate to describe the "ensemble behaviour" of the reference datasets and, therefore, it has been reported in the manuscript.

[Figure]

Fig. R3 : Taylor diagrams of the multiannual mean (1980-2005) of the DJFMA average snow water equivalent as described by climate models against the Multi-Reference-Median (MRM) calculated averaging NSIDC, CFSR, MERRA, ERAI/Land and 20CR climatologies. All datasets are projected onto the same reference grid at 0.7°lon.

Absence of spatial (geographical) analysis of the differences between the various spatial grids. To compare the grids you use 3 "quality" parameters reported in the Taylor diagrams. Even if Pearson-r is a measure of the association between variables and allows a global comparison of spatial patterns, I think that a spatial (or geographical) analysis of residuals (or the differences) is recommended to understand the effect of spatial localization. Doing so you should be able to improve the discussion of some climatological factors that are not integrated in the same in the same way in the models and related to air mass circulations for instance: North-South of the Alps – humid and cold air-mass flow from the North or East-West - continentality and humid air-mass flux coming from the Adriatic towards South Eastern Alpine and Pre-alpine domain) on eventual systematic and variable biases.. This spatial analysis should thus be done for some specific and well-chosen models.

Here the referee states that "To compare the grids, the spatial (or geographical) analysis of residuals (or the differences) is recommended", but actually this information is already contained in Figures 2,3,4 which show the differences between the various datasets (reanalysis, RCMs, GCMs) and the reference climatologies (EOBS for precipitation and temperature, NSIDC for snow water equivalent). Particularly important are, in our opinion, temperature and precipitation biases, that clearly show, for a given model, possible weaknesses related to the representation of the air mass circulation. These plots are commented in the corresponding section 4.1.1-4.1.3

**Some specific comments**

p. 1 l. 2. I'm very surprised about your conception of high spatial resolution in this abstract and in the whole text. For remote sensors (you use satellite data) hectometric and higher ground sampling distance corresponds to low and very low spatial resolution which don't allow any description of bio- geo-physical

processes on the Earth surface characterised by very high spatial frequency that are typical of mountainous area and especially the spatial variability of snow cover characteristics !

We agree that the definition of high resolution depends on the context. In the abstract P1 L2 ("high resolution, regional, observation-based gridded datasets"), "high resolution" refers to the typical spatial scales at which snow processes occur, i.e. less than 1 km. This has been clarified in the text (P1 L2) Later on, when speaking about the resolutions of global climate models, the concepts of "high" and "low" resolutions refer to the typical horizontal grid size of the state-of-the-art numerical climate models (CMIP5), ranging from 70 to 400 km. In this case "high resolution GCMs" are those with grid size equal or finer than 1.25° (about 125 km). We added in the introduction (P3 L3-5) a sentence to clarify these definitions.

p. 1 l. 20 "The shift of the 0_C isotherm to higher elevations ..." Is it demonstrated overall on the GAR ?

Yes, because of an overall increase of surface temperatures (see i.e. Gobiet et al., 2014; Hantel et al., 2012; Serquet et al 2011; Beniston, 2003). We added the references in the text at P2 L7

p. 1 l. 22 "...decrease in the solid-to-total precipitation ratio in low- and mid-altitude mountain areas." What do you mean by low and mid- altitude? Does that definition depends on the climatological sub-domain within the GAR ?

At this point we are presenting a general picture, not focused on the Alps, and with "low and mid-altitude" actually we intend "areas with temperatures closer to the melting point". We have better specified this in the text at P2 L5-8 , thank you.

p.3 l. 14 What do you mean by large scale? The notion of scale in your document is somehow perturbing for cartographers and geographers that are specifically doing multiscale spatial analysis (see also p. 3 l. 14, p. 16 l. 1 for instance)! A map with a scale of 1:10000 is a large scale map that allows the representation and analysis of local physical phenomenon with small autocorrelation distance (high spatial variability). At the contrary a map with a scale of 1:1000000 is a small scale map that allows the representation and analysis of global phenomenon.

This sentence was present in a preliminary version of the paper but it has been removed in the version published on the online TCD http://www.the-cryosphere-discuss.net/tc-2016-280/tc-2016-280.pdf. Interestingly, "large-scale" and "small-scale" have opposite meaning in cartography (as pointed out by the reviewer) and in climate/geophysical fluid dynamics, where the large scales are those with the largest spatial extent and the small scales are those with smaller spatial extent. Curious discrepancy (in fact, opposite meaning) of terms in two neighbouring fields of research.

p. 6 l. 31-32 "Global climate models, also the most spatially resolved ones, do not take into proper account elevations above 1500 m a.s.l. over the GAR."
It's really a critical issue because it seems that "a very weak increasing trend towards heavier snowfalls has persisted since the 1960s" until 1999 in the Swiss Alps above the altitude of 1300 m as demonstrated by LATERNSER and SCHNEEBELI (2003, DOI: 10.1002/joc.912), for instance. But this research emphases the snow cover extent using low spatial resolution AVHRR images and you correctly state that satellite products provide a reliable picture of snow cover extent which is not the case for snow depth or snow water equivalent (p. 2 l. 20 and 21).

Yes, moreover, the period over which those trends are calculated (1931-1999) does not consider the last 17 years, generally characterized by low snow.

p. 10 l. 2 "... arbitrarily chosen ..."
This is not an acceptable. You should provide a scientific justification!

We chose the ERA-Interim Land grid as it has intermediate resolution between RCM and GCM grids. This explanation has been added in the methodology.

p. 10 l. 9 "... a wider distribution of SNW values, ..."→"... a wider statistical dispersion of SNW values, ..." (if I understand correctly)

It has been corrected in the text, thank you.

p.10 l. 4 "This second approach allows to minimize the impact of the horizontal resolution on the performances of GCMs."
To Be Rewritten see next comment.

p. 10 l. 18 "... reduce the impact of the horizontal resolution on their performances ..."
I guess "their" refers to the models, then this sentence is not true. The impacts of the horizontal resolution on the models performance will not be reduced performing the reshaping of the reference datasets at the resolution of each GCM. To Be Rewritten.

Yes, we agree. We have changed the sentence (P18 L5-8) which now reads: "An alternative approach has been devised to provide a fair comparison of the GCMs. Each GCM is compared to the MRM after having conservatively remapped each reference dataset onto the individual GCM grid, so that the reference is reshaped each time according to the model resolution. This approach allows for a fair evaluation of the GCM at the model's grid, regardless of its resolution."
Thank you for the comment.

p. 10 l. 28 "... of at least 1.25° ..."→"... finer than 1.25° ..."
In the document the concept of resolution is confused with that of Ground Sampling Distance!

Thank you. We have corrected it in the manuscript. In climate models, "resolution" refers to the physical distance (meters or degrees) between two consecutive gridpoints, in latitudinal, longitudinal or vertical direction, on the grid used to compute the equations" (IPCC, 2013)

p. 15 l. 3 "... wet precipitation bias ..." Pleonasm! -→"... overestimated precipitation ..." or "... positive precipitation bias ..."

Corrected, thank you.

p. 16 l. 23-24 "At global scale, the spread over mountain regions has been estimated to be several times larger than over midlatitude regions (Mudryk et al., 2015)."
I don't understand why you compare midlatitude regions to mountain regions. The Alps are in a midlatitude region. You should complement the qualification of midlatitude regions!

Yes, we changed into "non-mountainous midlatitude regions", thank you

**Comments on the document's form (text, units, figures ...)**

Units must be controlled:

- - p. 3 l. 16 "~80° km spatial resolution"??

- - p. 8, l. 20 "105kg/m3 is not consistent with the unit used to describe the SNW in the reference

  datasets and the GCM (figure 2 for instance→kg/m2). It seems that you did this unit conversion to compute the mean annual cycle (figure 5 p. 13)

  p. 9 figure caption 2 "... with horizontal resolution higher than 1.25°."→"... with horizontal resolution finer than 1.25°"

  p. 9 figure 2 "Panels (j,k) report the multiannual mean of the DJFMA accumulated snowfall derived from the HISTALP dataset. You should give a precision about the unit. I guess the unit in mm refers to the water equivalent volume per area unit! This value could be expressed using the same unit than the reference datasets and the GCM (kg/m2) assuming that 1 mm corresponds to 1 l/m2 ~ 1 kg/m2 !

  p11. figure 3

- - Labels are not readable even for points corresponding to cases with large NSD !!
- - At that scale the large amount of points near the origin must be grouped in one class with a legend identifying all the point (dates or model) in the cluster.
- - Colour is not the best graphical variable to identify the signification of the point reported in the legend and it is probably not necessary. If points are grouped combine in one class and write the composition of the class. If points are dispersed then then label is sufficient!!!

All the above technical comments have been accepted and modified in the manuscript accordingly. Thank you.

---

## Author Comment (AC2) · 21 Apr 2017

**Reply to the comments by Reviewer#2 on the manuscript "Snow water equivalent in the Alps as seen by gridded datasets, CMIP5 and CORDEX climate models"**

We thank the Reviewer for the comments and suggestions regarding the discussion paper. We have addressed all the points he raised, performing supplementary analyses that in some cases were added in the manuscript or in the new Supplementary material. In brief, the new parts cover:

- The clarification of the paper objectives, including the motivation on why we kept the section on future projections.
- The evaluation of the RCMs in the historical period, previously based on ERA-Interim driven runs, and now extended also to the GCM-driven models. A new figure (Figure S03) was added in the Supplementary material.
- The analysis of the SNW distribution for different ranges of elevation, for all datasets (the references, GCMs, and RCMs). This analysis is now included in Fig. S04 of the Supplementary material.

Our point-by-point reply (black) to the suggestions and comments of the Reviewer (gray) is reported below.

Reviewer: "In this paper, the authors assess the snow water content in the Alps as represented in several atmospheric reanalyses, ERA-Interim-driven (and to a lower extent CMIP5-driven) regional atmosphere models (EURO-CORDEX), and numerous CMIP5 models. I appreciate the large amount of datasets analysed in this study, however I have several concerns with the paper in its current form, and I think that a major overhaul is required before publication.

First of all, the aim of the present study is not clearly stated. Does the paper aim to provide projections of snow water equivalent for end users (ecologists, road managers, ski resort), or does it aim to assess the models fidelity in order to point out limitations in our ability to project future snow water equivalent or for any other purpose? The aim should be better explained, and this should also be used to choose and justify which diagnostics are shown in this paper (e.g. why evaluating ERAinterin-driven RCMs in section 4.1 and 4.2 and additionally evaluating CMIP5-driven RCMs only in section 4.3?)."

Reply: The clarification of the aims of this paper was also provided as a response to Reviewer#1 to question #2. Now the objectives of the paper are clearly stated in the introduction (P3 L24 – P4 L6). This paper does not intend to deliver snow water equivalent projections for end users: without a proper *absolute* validation of the accuracy of the model, future projections would be pure speculation. Instead this paper aims to show and point out the strengths and limitations in the current knowledge of snow water equivalent characteristics at regional scale.

In brief the main objectives are:

- to assess the uncertainties in the characterisation of current snow water equivalent in the GAR, from both satellite/reanalyses and climate models.
- to explore how the current model uncertainties project into the future.

For the first objective we need to evaluate ERA-Interim-driven RCMs and, ideally, the AMIP simulations of the CMIP5 experiment, as pointed out by the referee (thank you for the suggestion). Nonetheless, out of

the 6 high resolution GCMs considered in this study, only two, CMCC-CM and MRI-CGCM3, have run AMIP simulations for the CMIP5 experiment (check in March 2017) and none of them is currently available for the download, apparently owing to issues with the servers. As of today, march 29th, we could not retrieve those data. At this stage it is impossible for us to evaluate the 2 AMIP runs.

For the second objective, we need GCM-driven RCMs and fully coupled GCMs. The scope of the manuscript is now better explained in the introduction.

"I also have a concern with the first diagnostics shown in this paper, i.e. the anomalies/biases represented on maps (Figs. 2-4). First, how are the datasets re-gridded prior to compute the difference? Furthermore, as the models (including reanalyses) miss the tail of the elevation distribution (as indicated in Fig.1), it is expected that they cannot account for high snowfalls observed in high-elevation areas. It seems to me that an alternative/complementary diagnostic would be to plot the snow water equivalent distribution per elevation bin. It would indicate whether the models behave well given their grid topography. I guess that the remapping used to build the Taylor diagram in Fig.5c partly addresses this, but it is not sufficient. In my opinion, this could replace sections 4.2 and 4.3 which I don't find very informative."

Throughout the paper the datasets are regridded using conservative remapping. This remapping allows the conservation of the quantity (SNW) from the original to the output grid.

Remapping methods do not change the original resolution of the datasets, so models and reanalyses that do not represent the tail of the elevation distribution are not expected to represent high snowfalls observed in high-elevation areas. As suggested by the reviewer we produced a plot representing the snow water equivalent distribution per elevation bin (Fig R4). Reanalyses represent elevations up to 2000-2500 m; CMIP5 models generally represent elevations up to 2000 m; RCMs describe high elevation areas up to 3000 m. This plot clarifies what elevation ranges are represented in each dataset, thus it has been included in the text in Figure S04. However this analysis does not show how close the modelled and the reference SNW patterns are, in terms of point-by-point correlation, mean error and variance. This information is instead given in the Taylor diagrams, which in our opinion provide much information in a concise way and, in our opinion, they cannot be replaced by the plot of SNW per elevation ranges alone.

[Figure]

Fig R4. Multiannual mean DJFMA SNW in the Greater Alpine Region spatially averaged over different elevation ranges (500 m wide). The elevation is derived from the topography of each model (reanalysis).

"In addition, despite the limitations mentioned for snow water equivalent derived from passive microwave satellite observations, no attempt is made to discuss the validity of such products. The minimum would be to compare the two datasets described in section 2.1 over their common period. Of course, comparing to more datasets or to in-situ measurements would be even better. Is there any evidence that these satellite datasets are more reliable than the other datasets?"

The two satellite datasets AMSRE and Global SWE have been compared over their common period (2003-2007) and the plot has been integrated in Fig. 2f. We thank the reviewer for this suggestion. Given the relative short period of overlapping (5 years) we did not investigate further the time series, but we reported and discussed previous validation papers (Section 2.1 P5 L10-16)

"I have a possible concern with the choice of the simulations presented in the manuscript. ERAinterin-driven RCMs are more similar to AMIP models (atmospheric- only GCMs driven by observed SSTs) than to CMIP models, so in section 4.1 and 4.2, I think that comparing AMIP GCMS to ERAinterin-driven RCMs would make more sense. Then, in section 4.3 and 4.4 where the CMIP5-driven RCMs are evaluated, it makes more sense to compare to CMIP5 models."

As stated before AMIP simulations are provided for only 2 HiRes GCMs and they are currently not available for download. However, we added Fig S03 in the supplementary material presenting the biases of GCM-driven RCM, to be compared to fully coupled CMIP5 models

"I have several other specific comments:

- Abstract, l.11: replace "latest" with "fifth" (in a couple of years, latest won't be clear)."

Done, thank you.

- 2nd and 3rd paragraph of the Introduction: there are also concerns related to snow itself (road & airport safety, ski resorts, . . .).

Thank you. We have mentioned in the text the impacts on winter tourism and we added 2 citations (Beniston et al., 2011, Rixen et al., 2011). We preferred not to mention airport and road safety because it is more related to extreme events, i.e. to temporal scales not covered by our analysis.

- Intro, l.25-26: "at relatively high spatial resolution" -> subjective, indicate a typical range.

Done, thank you.

- It would be interesting to discuss the reliability of satellite datasets in the Introduction. Note that the GlobSnow dataset is derived from satellite measurements but uses ground-based weather station data in the SWE retrieval.

Yes, we added:
- a discussion on reliability of satellite datasets in (Section 2.1 P5 L10-16)
- the fact that GlobSnow is based also on "surface measurements" (P3 L17). Thank you

- Section 2.3: Sabin et al. (2013) use LMDz as an atmosphere-only model (i.e. not coupled to an ocean), I don't know how relevant this is to the CMIP models.

Actually also Davini et al 2017 does. At present state of the art, ultra-high resolution simulations are AMIP only.

- Section 2.3, last paragraph, remove "at ISAC-CNR".
- Tab. 1: there should probably be a line between satellite products and reanalyses.

Done, thank you.

- Given that LMD is mentioned, I'm surprised not to see the IPSL models in the long CMIP5 list, but anyway, there are clearly enough models in this paper.

IPSL models provide the snow depth variable but not snow water equivalent. Being the focus of this study on the latter variable, IPSL models do not appear in the paper.

- Sections 2.3 and 2.4: mention what kind of outputs are used (daily means or monthly means?).

We used monthly means. This detail has been added in the text (P7 L18 and L32)

- Section 2.4: what is a "non-reliable snow accumulation trends"? (and what is a reliable trend?).

Pixels masked as "glaciers" do not reproduce the snowpack evolution (accumulation and melting) but they continuously accumulate snowfall in time (without melting). "Non-reliable trend" refers to this behaviour and it has been clarified in the text (P8 L1-5).

- Section 2.5, about "The ability of climate models to properly reproduce snow water equivalent depends both on the accuracy of their snow schemes and on the reliability of the atmospheric forcings": it actually depends on many kinds on biases in the regional model (e.g. radiation scheme, boundary-layer scheme, etc, all being able to eventually impact snow).

Yes, we see your point. Actually with "reliability of the forcings" we already include all possible biases due to the land-surface and atmospheric schemes. We have rephrased the sentence "The ability of climate models to properly reproduce snow water equivalent depends on the accuracy of their surface snow schemes and on the reliability of the atmospheric fields forcing the snowpack processes."

- Section 2.5: what is the interpolation method for HISTALP and EOBS?

EOBS is kept at its original resolution (0.25° lat-lon regular grid). HISTALP has been conservatively remapped to EOBS grid, as all the other datasets, for the comparison in Figure 2. This has been explained in the corresponding Section 4.1.1

- Section 3 could probably be merged with section 2 into a "datasets and methods" section.

Actually we prefer to keep them separate to make the text more readable.

- Fig.2: why showing the relative precipitation bias (in %) while the temperature and snow biases are shown as absolute errors?

Mainly to be consistent with a previous study by Kotlarsky et al., 2014, presenting the same maps for the same models over the full EURO-CORDEX domain. Here we present a focus on the Alpine region.

- Fig.2: the caption "snow water equivalent in the EOBS observational dataset and the NSIDC Global Monthly EASE-Grid Snow Water Equivalent Climatology respectively" is misleading, it would be clearer at a first read to write that EOBS relates to (a) and (b) while NSIDC relates to (c).

Done, thank you.

- Section 4.1.1, about "In order to facilitate the comparison we present the differences with respect to a given dataset: the NSIDC Global SNW Climatology for SNW, since it is available for a longer period (1980-2005) than the other satellite product AMSR-E (2003-2011)". Ok, but it is a pity not to compare these two products aver the common period, especially given that you have claimed that "we expand the study by Mudryk et al. (2015) by including additional global SNW gridded datasets obtained from remote sensing" in the Introduction.

As previously mentioned, we have compared the two SNW satellite datasets over their common period (2003-2007) and the results are reported in Fig 2f. Given the relative short period of overlapping (5 years) we did not investigate further the time series, but we presented and discussed two papers on the validation of the two satellite products (Section 2.1 P5 L10-16). Thank you for the suggestion.

- Section 4.1.2: I would not say that REMO2009 is much better than the other RCMs, there is a substantial warm bias all over the domain (except maybe just over the mountain range) that could explain the relatively lower bias in SNW compared to other RCMs. Also, I would replace "CCLM4-8-17 and REMO2009 models which present no issues" with "CCLM4-8-17 and REMO2009 models which present weaker biases than other RCMs".

We agree that the performance of REMO2009 are comparable to other RCMs (please note that the plot in Fig 4m has been updated after finding an error in the computation of the DJFMA mean). We have better explained in the text (Sections 2.4 and 4.1.3) the "issues" in ALADIN53, HIRAM5 and RACMO22E models: "in glacier-masked pixels they show continuous snow accumulation and no melting. As this feature hampers the regridding of the models and the calculation of spatial averages over the GAR" we did not consider them for investigating the annual cycle and its projected changes at mountain range scale.

- Section 4.1.3: what period is used for the CMIP5 models, 1980-2005 or 1850-2005?

1980-2005, as clarified at P13 L4-5.

- Section 4.2 and its Taylor diagrams. I don't find the spatial correlation very relevant here, because it mostly relies on correlations between the topographies. Similar comment for RMSE and NSD.

We agree that the correlation coefficient R mainly reflects the *model* topography but we do not this this is a limitation because each model has its own topography, at its own resolution. It would have been meaningless if all models were using the same topography. In our case the objective is to measure the similarity between climate model climatologies (provided at different resolution) and a reference pattern. In such case RMSE, NSD and R provide, in our opinion, a good measure of this similarity.

- Why removing the worst RCMs in section 4.2?

We did not remove the worst RCMs but the models presenting pixels characterized by continuous snow accumulation and no melting, possibly areas masked as glaciers. As this feature hampers the regridding of the models and the calculation of spatial averages over the GAR we retained only two RCMs out of the five for further investigation. This has been explained in the text at P8 L1-5

- I am a bit lost, why using CMIP5-driven RCMs to analyse the seasonal cycle in section 4.3 and not to evaluate the mean spatial patterns in sections 4.1 and 4.2?

Yes, we added the evaluation of the GCM-driven RCMs in section 4.1.3, with one additional plot (Figure S3) in the Supplementary material.

- Section 4.3: I would not call 20CR a "reference dataset", it is a coarse atmospheric GCM only constrained by surface pressure and SSTs, probably more comparable to a coarse AMIP model. . .

Thank you for this suggestion. As already mentioned in the Response to Reviewer 1, in the revised version of the manuscript the 20CR reanalysis is not considered as a "reference" any longer. In fact, we repeated all the analyses excluding the 20CR one from the Multi-Reference-Mean, and consequently figures 5, 6 and 7 have been updated in the main text. The new procedure is explained at lines P12L31-34.

---

## Referee Report (RR1)

**Snow water equivalent in the Alps as seen by gridded datasets, CMIP5 and CORDEX climate models**

*Silvia Terzago, Jost von Hardenberg, Elisa Palazzi, and Antonello Provenzale*

**Second review by reviewer 1, Yves CORNET.**

The last version of the document has been strongly improved. I congratulate the authors. I'm globally very satisfied of the complete and pertinent answers given to my question and comments in the paper and by interesting supplementary material provided.

Regarding your comment *"We noted that for few comments of Revision 1, the page and lines indicated by the Reviewer do not match exactly the page/lines in the manuscript published on TCD."* Indeed I recognize I had some difficulties to manage the web access to the documents!!! I read the first version of your paper dated 6 December 2016 and not the version dated 23 January 2017. This had some consequences (see below) because figure 2 and 3 of the version dated 23 January is much more explicit than the homologous figures of the version dated 6 December. Sorry!!!

**Answers to the sections**
**"Methodological issues"**
**"Research paradigms and hypothesis to be demonstrated"**

I still remain skeptical about the SWE's prediction. But the new version of the paper is presented with more information and precautions so that the reader correctly evaluates these predictions in terms of magnitude of variation and inconsistencies.

I recognize my misunderstanding of the weighting procedure used to aggregate spatially over the GAR. In the last version of your paper the explanation is now clear. I think you had to clarify it.

Regarding remapping the conservative procedure is now specified, so my doubts are blurred.

As suggested, you have now canceled of the 20CR reanalysis in the mean reference computation as suggested. I'm convinced it was justified.

I remain a little bit disappointed about the answer given regarding the effect of the interdependency between Global SWE Climatology and CFSR snow outputs. Their weight remains high in the computation of aggregated value. It is not a question of similar range of variation!! But In the last explanation you have provided it is clear that you have a very good knowledge of those products (much better than mine). So I trust you …

Thank you for the supplementary material and the precision given to get ensemble mean of HiRes models (MMM-HiRes) using only one of the 3 CESM models that show similar spatial patterns. This should reduce the non-independency effect. Regarding GCMs It is obvious that no further analysis is needed.

You provide an interesting investigation of two approaches that provide more consistent results. Fig. R3 and 5a are explicit.

My question about the usefulness of spatial analysis of differences between two gridded products to compare them in complement of only using Pearson-r association coefficient was founded by the fact that similar values of r values can hide very different spatial patterns of consistency/inconsistency. In the case of random spatial distribution of difference values the absence of geographical and climatological reasons can be assumed. In the case of high spatial autocorrelation of differences, a geographical/climatological must obviously be investigated. As I read the 6 December version of your paper (see above), my question about spatial analysis was justified. The new version of figures 2 and 3 (already present in the 23 January version) provide new insight that answer in a totally satisfying way to my question. Air temperature and precipitation are indeed the output of models that can explain spatial variation of SWE biases. So sorry again for my review of the oldest version of your paper!

**Reactions to the sections**
**"Some specific comments"**
**"Comments on the document's form (text, units, figures …)"**

I agree with all the reactions to the specific comments.

All the requested modification of from have been applied.

**Conclusion**

I suggest the editor to purplish this version of the document.

*Concittadini (per meta`) e collegiali complimenti !*

```
Yves CORNET,
Prof. of Remote Sensing and Photogrammetry
Geomatics Unit (University of Liège)

Institut de Physique (B5a)
Allée du 6 Août, 17
B-4000 Liège (Belgium)

Tél. +32 04 366 5371
Fax. +32 04 366 9693

http://www.geo.ulg.ac.be/
http://www.Dept-geo.ulg.ac.be/
```

---

## Author Response (AR2)

Dear Editor,

we are pleased to send you enclosed the final version of our manuscript entitled "Snow water equivalent in the Alps as seen by gridded datasets, CMIP5 and CORDEX climate models".

In this version the minor revisions requested by the Reviewers (correction of a typo highlighted by Reviewer #2) have been addressed.

We would like to thank you for your suggestions after the initial paper submission. We are truly grateful to the two Reviewers for their constructive comments, their advices helped us to significantly improve the paper.

With kind regards
Silvia Terzago and co-authors